# Efficient Meta Reinforcement Learning for Preference-based Fast Adaptation

**Zhizhou Ren**[12], **Anji Liu**[3], **Yitao Liang**[45], **Jian Peng**[126], **Jianzhu Ma**[6]
[1]Helixon Ltd. [2]University of Illinois at Urbana-Champaign
[3]University of California, Los Angeles
[4]Institute for Artificial Intelligence, Peking University
[5]Beijing Institute for General Artificial Intelligence
[6]Institute for AI Industry Research, Tsinghua University
zhizhour@helixon.com, liuanji@cs.ucla.edu
yitaol@pku.edu.cn, jianpeng@illinois.edu
majianzhu@tsinghua.edu.cn

## Abstract

Learning new task-specific skills from a few trials is a fundamental challenge for artificial intelligence. Meta reinforcement learning (meta-RL) tackles this problem by learning transferable policies that support few-shot adaptation to unseen tasks. Despite recent advances in meta-RL, most existing methods require the access to the environmental reward function of new tasks to infer the task objective, which is not realistic in many practical applications. To bridge this gap, we study the problem of few-shot adaptation in the context of human-in-the-loop reinforcement learning. We develop a meta-RL algorithm that enables fast policy adaptation with preference-based feedback. The agent can adapt to new tasks by querying human's preference between behavior trajectories instead of using per-step numeric rewards. By extending techniques from information theory, our approach can design query sequences to maximize the information gain from human interactions while tolerating the inherent error of non-expert human oracle. In experiments, we extensively evaluate our method, *Adaptation with Noisy OracLE* (ANOLE), on a variety of meta-RL benchmark tasks and demonstrate substantial improvement over baseline algorithms in terms of both feedback efficiency and error tolerance.

## 1 Introduction

Reinforcement learning (RL) has achieved great success in learning complex behaviors and strategies in a variety of sequential decision-making problems, including Atari games (Mnih et al., 2015), board game Go (Silver et al., 2016), MOBA games (Berner et al., 2019), and real-time strategy games (Vinyals et al., 2019). However, most these breakthrough accomplishments are limited in simulation environments due to the sample inefficiency of RL algorithms. Training a policy using deep reinforcement learning usually requires millions of interaction samples with the environment, which is not practicable in real-world applications. One promising methodology towards breaking this practical barrier is *meta reinforcement learning* (meta-RL Finn et al., 2017). The goal of meta-RL is to enable fast policy adaptation to unseen tasks with a small amount of samples. Such an ability of few-shot adaptation is supported by meta-training on a suite of tasks drawn from a prior task distribution. Meta-RL algorithms can extract transferable knowledge from the meta-training experiences by exploiting the common structures among the prior training tasks. A series of recent works have been developed to improve the efficiency of meta-RL in several aspects, *e.g.*, using off-policy techniques to improve the sample efficiency of meta-training (Rakelly et al., 2019; Fakoor et al., 2020), and refining the exploration-exploitation trade-off during adaptation (Zintgraf et al.,

36th Conference on Neural Information Processing Systems (NeurIPS 2022).

2020; Liu et al., 2021). There are also theoretical works studying the notion of few-shot adaptation and knowledge reuse in RL problems (Brunskill and Li, 2014; Wang et al., 2020; Chua et al., 2021).

While recent advances remarkably improve the sample efficiency of meta-RL algorithms, little work has been done regarding the type of supervision signals adopted for policy adaptation. The adaptation procedure of most meta-RL algorithms is implemented by either fine-tuning policies using new trajectories (Finn et al., 2017; Fakoor et al., 2020) or inferring task objective from new reward functions (Duan et al., 2016; Rakelly et al., 2019; Zintgraf et al., 2020), both of which require the access to the environmental reward function in new tasks to perform adaptation. Such a reward-driven adaptation protocol may become impracticable in many application scenarios. For instance, when a meta-trained robot is designed to provide customizable services for non-expert human users (Prewett et al., 2010), it is not realistic for the meta-RL algorithm to obtain per-step reward signals directly from the user interface. The design of reward function is a long-lasting challenge for reinforcement learning (Sorg et al., 2010), and there is no general solution to specify rewards for a particular goal (Amodei et al., 2016; Abel et al., 2021). Existing techniques can support reward-free meta-training by constructing a diverse policy family through unsupervised reward generation (Gupta et al., 2020), and reward-free few-shot policy adaptation remains a challenge (Liu et al., 2020).

In this paper, we study the interpolation of meta-RL and human-in-the-loop RL, towards expanding the applicability of meta-RL in practical scenarios without environmental rewards. We pursue the few-shot adaptation from human preferences (Fürnkranz et al., 2012), where the agent infers the objective of new tasks by querying a human oracle to compare the preference between pairs of behavior trajectories. Such a preference-based supervision is more intuitive than numeric rewards for human users to instruct the meta policy. *e.g.*, the human user can watch the videos of two behavior trajectories and label a preference order to express the task objective (Christiano et al., 2017). The primary goal of such a preference-based adaptation setting is to suit the user's preference with only a few preference queries to the human oracle. Minimizing the burden of interactions is a central objective of human-in-the-loop learning. In addition, we require the adaptation algorithm to be robust when the oracle feedback carries some noises. The human preference is known to have some extent of inconsistency (Loewenstein and Prelec, 1992), and the human user may also make unintentional mistakes when responding to preference queries. The tolerance to feedback error is an important evaluation metric for preference-based reinforcement learning (Lee et al., 2021a).

To develop an efficient and robust adaptation algorithm, we draw a connection with a classical problem called *Rényi-Ulam's game* (Rényi, 1961; Ulam, 1976) from information theory. We model the preference-based task inference as a noisy-channel communication problem, where the agent communicates with the human oracle and infers the task objective from the noisy binary feedback. Through this problem transformation, we extend an information quantification called *Berlekamp's volume* (Berlekamp, 1968) to measure the uncertainty of noisy preference feedback. This powerful toolkit enables us to design the query contents to maximize the information gain from the noisy preference oracle. We implement our few-shot adaptation algorithm, called *Adaptation with Noisy OracLE* (ANOLE), upon an advanced framework of probabilistic meta-RL (Rakelly et al., 2019) and conduct extensive evaluation on a suite of Meta-RL benchmark tasks. The experiment results show that our method can effectively infer the task objective from limited interactions with the preference oracle and, meanwhile, demonstrate robust error tolerance against the noisy feedback.

## 2 Preliminaries

### 2.1 Problem Formulation of Meta Reinforcement Learning

**Task Distribution.** The same as previous meta-RL settings (Rakelly et al., 2019), we consider a task distribution $p(\mathcal{T})$ where each task instance $\mathcal{T}$ induces a *Markov Decision Process* (MDP). We use a tuple $\mathcal{M}_{\mathcal{T}} = \langle \mathcal{S}, \mathcal{A}, T, R_{\mathcal{T}} \rangle$ to denote the MDP specified by task $\mathcal{T}$. In this paper, we assume task $\mathcal{T}$ does not vary the environment configuration, including state space $\mathcal{S}$, action space $\mathcal{A}$, and transition function $T(s' \mid s, a)$. Only reward function $R_{\mathcal{T}}(\cdot \mid s, a)$ conditions on task $\mathcal{T}$, which defines the agent's objectives for task-specific goals. The meta-RL algorithm is allowed to sample a suite of training tasks $\{\mathcal{T}_i\}_{i=1}^{N}$ from the task distribution $p(\mathcal{T})$ to support meta-training. In the meta-testing phase, a new set of tasks are drawn from $p(\mathcal{T})$ to evaluate the adaptation performance.

**Preference-based Adaptation.** We study the few-shot adaptation with preference-based feedback, in which the agent interacts with a black-box preference oracle $\Omega_{\mathcal{T}}$ rather than directly receiving task-specific rewards from the environment $\mathcal{M}_{\mathcal{T}}$ for meta-testing adaptation. More specifically, the agent can query the preference order of a pair of trajectories $\langle \tau^{(1)}, \tau^{(2)} \rangle$ through the black-box oracle $\Omega_{\mathcal{T}}$. Each trajectory is a sequence of observed states and agent actions, denoted by $\tau = \langle s_0, a_0, s_1, a_1, \cdots, s_L \rangle$ where $L$ is the trajectory length. For each query trajectory pair $\langle \tau^{(1)}, \tau^{(2)} \rangle$, the preference oracle $\Omega_{\mathcal{T}}$ would return either $\tau^{(1)} \succ \tau^{(2)}$ or $\tau^{(1)} \prec \tau^{(2)}$ according to the task specification $\mathcal{T}$, where $\tau^{(1)} \succ \tau^{(2)}$ means the oracle prefers trajectory $\tau^{(1)}$ to trajectory $\tau^{(2)}$ under the context of task $\mathcal{T}$. When the preference orders of $\tau^{(1)}$ and $\tau^{(2)}$ are equal, both returns are valid. The preference-based adaptation is a weak-supervision setting in comparison with previous meta-RL works using per-step reward signals for few-shot adaptation (Finn et al., 2017; Rakelly et al., 2019). The purpose of this adaptation setting is to simulate the practical scenarios with human-in-the-loop supervision (Wirth et al., 2017; Christiano et al., 2017). We consider two aspects to evaluate the ability of an adaptation algorithm:

- **Feedback Efficiency.** The central goal of meta-RL is conducting fast adaptation to unseen tasks with a few task-specific feedback. We consider the adaptation efficiency in terms of the number of preference queries to oracle $\Omega_{\mathcal{T}}$. This objective expresses the practical demand that we aim to reduce the burden of preference oracle since human feedback is expensive.

- **Error Tolerance.** Another performance metric is on the robustness against the noisy oracle feedback. The preference oracle $\Omega_{\mathcal{T}}$ may carry errors in practice, *e.g.*, the human oracle may misunderstand the query message and give wrong feedback. Tolerating such oracle errors is a practical challenge for preference-based reinforcement learning.

## 2.2 Meta Reinforcement Learning with Probabilistic Task Embedding

**Latent Task Embedding.** We follow the algorithmic framework of *Probabilistic Embeddings for Actor-critic RL* (PEARL; Rakelly et al., 2019). The task specification $\mathcal{T}$ is modeled by a latent task variable (or latent task embedding) $z \in \mathcal{Z} = \mathbb{R}^d$ where $d$ denotes the dimension of the latent space. With this formulation, the overall paradigm of the meta-training procedure resembles a multi-task RL algorithm. Both policy $\pi(a|s; z)$ and value function $Q(s, a; z)$ condition on the latent task variable $z$ so that the representation of $z$ can be end-to-end learned with the RL objective to distinguish different task specifications. During meta-testing, the adaptation is performed in the low-dimensional task embedding space rather than the high-dimensional parameter space.

**Adaptation via Probabilistic Inference.** To infer the task embedding $z$ from the latent task space, PEARL trains an inference network (or context encoder) $q(z|\mathbf{c})$ where $\mathbf{c}$ is the context information including agent actions, observations, and rewards. The output of $q(z|\mathbf{c})$ is probabilistic, *i.e.*, the agent has a probabilistic belief over the latent task space $\mathcal{Z}$ based on its observations and received rewards. We use $q(z)$ to denote the prior distribution for $\mathbf{c} = \emptyset$. The adaptation protocol of PEARL follows the framework of *posterior sampling* (Strens, 2000). The agent continually updates its belief by interacting with the environment and refine its policy according to the belief state. This algorithmic framework is generalized from Bayesian inference (Thompson, 1933) and has solid background in reinforcement learning theory (Agrawal and Goyal, 2012; Osband et al., 2013; Russo and Van Roy, 2014). However, some recent works show that the empirical performance of neural inference networks highly rely on the access to a dense reward function (Zhang et al., 2021; Hua et al., 2021). When the task-specific reward signals are sparsely distributed along the agent trajectory, the task inference given by context encoder $q(z|\mathbf{c})$ would suffer from low sample efficiency and cannot accurately decode task specification. This issue may worsen in our adaptation setting since only trajectory-wise preference comparisons are available to the agent. It motivates us to explore new methodology for few-shot adaptation beyond classical approaches based on posterior sampling.

## 3 Preference-based Fast Adaptation with A Noisy Oracle

In this section, we will introduce our method, *Adaptation with Noisy OracLE* (ANOLE), a novel task inference algorithm for preference-based fast adaptation. The goal of our approach is to achieve both high feedback efficiency and error tolerance.

### 3.1 Connecting Preference-based Task Inference with Rényi-Ulam's Game

To give an information-theoretic view on the task inference problem with preference feedback, we connect our problem setting with a classical problem called *Rényi-Ulam's Game* (Rényi, 1961; Ulam, 1976) from information theory to study the interactive learning procedure with a noisy oracle.

**Definition 1** (Rényi-Ulam's Game). *There are two players, called A and B, participating the game. Player A thinks of something in a predefined element universe, and player B would like to guess it. To extract information, player B can ask some questions to player A, and the answers to these questions are restricted to "yes/no". A given percentage of **player A's answers can be wrong**, which **requires player B's question strategy to be error-tolerant**.*

In the literature of information theory, Rényi-Ulam's game specified in Definition 1 is developed to study the error-tolerant communication protocol for noisy channels (Shannon, 1956; Rényi and Makkai-Bencsáth, 1984). Most previous works on Rényi-Ulam's game focus on the error tolerance of player B's question strategy, *i.e.*, how to design the question sequence to maximize the information gain from the noisy feedback (Pelc, 2002). In this paper, we consider the online setting of Rényi-Ulam's game, where player B is allowed to continually propose queries based on previous feedback.

We draw a connection between Rényi-Ulam's game and preference-based task inference. In the context of preference-based meta adaptation, the task inference algorithm corresponds to the questioner player B, and the preference oracle $\Omega$ plays the role of responder player A. The preference feedback given by oracle $\Omega$ is a binary signal regarding the comparison between two trajectories, which has the same form as the "yes/no" feedback in Rényi-Ulam's game. The goal of the task inference algorithm is to search for the true task specification in the task space using minimum number of preference queries while tolerating the errors in oracle feedback. The similarity in problem structures motivates us to extend techniques from Rényi-Ulam's game to preference-based task inference.

### 3.2 An Algorithmic Framework for Preference-based Fast Adaptation

In this section, we discuss how we transform the preference-based task inference problem to Rényi-Ulam's game and introduce the basic algorithmic framework of our approach to perform few-shot adaptation with a preference oracle.

**Transformation to Rényi-Ulam's Game.** The key step of the problem transformation is establishing the correspondence between the preference query in preference-based task inference and the natural-language-based questions in Rényi-Ulam's game. In classical solutions of Rényi-Ulam's game, a general format of player B's questions is to ask whether the element in player A's mind belongs to a certain subset of the element universe (Pelc, 2002), whereas the counterpart of question format in preference-based task inference is restricted to querying whether the oracle prefers one trajectory to another. To bridge this gap, we use a model-based approach to connect the oracle preference feedback with the latent space of task embeddings. We train a preference predictor $f_\psi(\cdot; z)$ that predicts the oracle preference according to the task embedding $z$. This preference predictor can transform each oracle preference feedback to a separation on the latent task space, *i.e.*, the preference prediction $f_\psi(\cdot; z)$ given by task embeddings in a subspace of $z$ can match the oracle feedback, and the task embeddings in the complementary subspace lead to wrong predictions. Through this transformation, the task inference algorithm can convert the binary preference feedback to the assessment of a subspace of latent task embeddings, which works in the same mechanism as previous solutions to Rényi-Ulam's game.

To realize the problem transformation, we consider a simple and direct implementation of the preference predictor. We train $f_\psi(\cdot; z)$ on the meta-training tasks by optimizing the preference loss:

$$\mathcal{L}^{\text{Pref}}(\psi) = \mathbb{E}\left[D_{\text{KL}}\left(\mathbb{I}\big[\tau^{(1)} \succ \tau^{(2)} \mid \mathcal{T}\big] \,\Big\|\, f_\psi\big(\tau^{(1)} \succ \tau^{(2)}; z\big)\right)\right], \tag{1}$$

where $\psi$ denotes the parameters of the preference predictor, $D_{\text{KL}}(\cdot\|\cdot)$ denotes the Kullback–Leibler divergence, and $\mathbb{I}[\tau^{(1)} \succ \tau^{(2)} \mid \mathcal{T}]$ is the ground-truth preference order specified by the task specification $\mathcal{T}$ (*e.g.*, specified by the reward function on training tasks). The trajectory pair $\langle \tau^{(1)}, \tau^{(2)} \rangle$ is drawn from the experience buffer, and $z$ is the task embedding vector encoding $\mathcal{T}$. More implementation details are included in Appendix B.2.

**Basic Algorithmic Framework.** To facilitate discussions, we introduce some notations to model the interaction with the preference oracle $\Omega_{\mathcal{T}}$. Suppose the adaptation budget supports $K$ online preference queries to oracle $\Omega_{\mathcal{T}}$, which divides this interactive procedure into $K$ rounds. We define the notation of query context set as Definition 2 to represent the context information extracted from the preference queries during the online oracle interaction.

**Definition 2** (Query Context Set). *The query context set $\mathcal{Q}_k = \{\langle \tau_j^{(1)} \succ \tau_j^{(2)} \rangle\}_{j=1}^k$ denotes the set of preference queries completed at the first $k$ rounds, in which the task inference protocol queries the preference order between the trajectory pair $\langle \tau_j^{(1)}, \tau_j^{(2)} \rangle$ at the $j$th round. To simplify the notations, the trajectory pair $\langle \tau_j^{(1)}, \tau_j^{(2)} \rangle$ are relabeled according to the oracle feedback so that the preference order given by oracle $\Omega$ is $\tau_j^{(1)} \succ \tau_j^{(2)}$ for any $1 \le j \le k$.*

The query context set $\mathcal{Q}_k$ concludes the context information obtained from the oracle $\Omega$ in the first $k$ rounds. After completing the query at round $k$, the task inference algorithm needs to decide the next-round query trajectory pair $\langle \tau_{k+1}^{(1)}, \tau_{k+1}^{(2)} \rangle$ based on the context information stored in $\mathcal{Q}_k$. By leveraging the model-based problem transformation to Rényi-Ulam's game, we can assess the quality of a task embedding $z$ by counting the number of mismatches with respect to oracle preferences, denoted by $\mathcal{E}(z; \mathcal{Q}_k)$:

$$\mathcal{E}(z; \mathcal{Q}_k) = \sum_{(\tau^{(1)} \succ \tau^{(2)}) \in \mathcal{Q}_k} \mathbb{I}\left[ f_\psi(\tau^{(1)} \succ \tau^{(2)}; z) < f_\psi(\tau^{(2)} \succ \tau^{(1)}; z) \right]. \tag{2}$$

The overall algorithmic framework of our preference-based few-shot adaptation method, *Adaptation with Noisy OracLE* (ANOLE), is summarized in Algorithm 1.

---

**Algorithm 1** Adaptation with Noisy OracLE (ANOLE)

---

1: **input:** a preference oracle $\Omega_{\mathcal{T}}$, the budget of oracle queries $K$
        a prior task distribution $q(z)$, the size of candidate pool $M$
        a query generation protocol GENERATEQUERY$(\cdot)$
2: $\widehat{Z} \leftarrow \{z_j \sim q(z)\}_{j=1}^M$                                    ▷ sample a candidate pool
3: $\mathcal{Q}_0 \leftarrow \emptyset$                                            ▷ initialize context information
4: **for** $k = 1$ **to** $K$ **do**
5:     $\langle \tau_k^{(1)}, \tau_k^{(2)} \rangle \leftarrow$ GENERATEQUERY$(\widehat{Z}, \mathcal{Q}_{k-1})$    ▷ **critical step:** query generation (Eq. (6))
6:     $\langle \tau_k^{(u)} \succ \tau_k^{(v)} \rangle \leftarrow \Omega_{\mathcal{T}}(\tau_j^{(1)}, \tau_j^{(2)})$ where $u, v \in \{1, 2\}$     ▷ request oracle feedback
7:     $\mathcal{Q}_k \leftarrow \mathcal{Q}_{k-1} \cup \{\langle \tau_k^{(u)} \succ \tau_k^{(v)} \rangle\}$             ▷ update context information
8: **return:** $\arg\min_{z \in \widehat{Z}} \mathcal{E}(z; \mathcal{Q}_K)$

---

The first step of our task inference algorithm is sampling a candidate pool $\widehat{Z}$ of latent task embeddings from the prior distribution $q(z)$. Then we perform $K$ rounds of candidate selection by querying the preference oracle $\Omega_{\mathcal{T}}$. The design of the query generation protocol GENERATEQUERY$(\cdot)$ is a critical component of our task inference algorithm and will be introduced in section 3.3. The final decision would be the task embedding with minimum mismatch with the oracle preference, *i.e.*, $\arg\min_{z \in \widehat{Z}} \mathcal{E}(z; \mathcal{Q}_K)$.

### 3.3 Error-Tolerant Task Inference for Noisy Preference Oracle

The problem transformation to Rényi-Ulam's game enables us to leverage techniques from information theory to develop query strategy with both high feedback efficiency and error tolerance.

**Binary-Search Paradigm.** The basic idea is conducting a binary-search-like protocol to leverage the binary structure of preference feedback: *After each round of oracle interaction, we shrink the candidate pool $\widehat{Z}$ by removing those task embeddings leading to wrong preference predictions $f_\psi(\cdot; z)$.* An ideal implementation of such a binary-search protocol with noiseless feedback is expected to roughly eliminate half of candidates using each single oracle preference feedback, which achieves the information-theoretic lower bound of interaction costs. In practice, we pursue to handle noisy

feedback, since both the preference oracle $\Omega_{\mathcal{T}}$ and the preference predictor $f_\psi(\cdot; z)$ may carry errors. An error-tolerant binary-search protocol requires to establish an information quantification (*e.g.*, an uncertainty metric) to evaluate the information gain of each noisy oracle feedback. The goal of oracle interaction is to rapidly reduce the uncertainty of task inference rather than simply eliminate the erroneous candidates. In this paper, we extend an information-theoretic tool called *Berlekamp's volume* (Berlekamp, 1968) to develop such an uncertainty quantification.

**Berlekamp's Volume.** One classical tool to deal with erroneous information in search problems is *Berlekamp's volume*, which is first proposed by Berlekamp (1968) and has been explored by subsequent works in numerous variants of Rényi-Ulam's game (Rivest et al., 1980; Pelc, 1987; Lawler and Sarkissian, 1995; Aigner, 1996; Cicalese and Deppe, 2007). The primary purpose of Berlekamp's volume is to mimic the notion of Shannon entropy from information theory (Shannon, 1948) and specialize in the applications to noisy-channel communication (Shannon, 1956) and error-tolerant search (Rényi, 1961). We refer readers to Pelc (2002) for a comprehensive literature review of the applications of Berlekamp's Volume and the solutions to Rényi-Ulam's game.

In Definition 3, we rearrange the definition of Berlekamp's volume to suit the formulation of preference-based learning.

**Definition 3** (Berlekamp's Volume). *Suppose the budget supports $K$ oracle queries in total, and the oracle may have at most $K_E$ incorrect feedback among these queries. Berlekamp's volume of a query context set $\mathcal{Q}_k$ is defined as follows:*

$$\mathcal{BVol}_{\widehat{Z}}(\mathcal{Q}_k) = \sum_{z \in \widehat{Z}} vol_z(\mathcal{Q}_k), \qquad vol_z(\mathcal{Q}_k) = \sum_{\ell=0}^{K_E - \mathcal{E}(z; \mathcal{Q}_{k-1})} \binom{K-k}{\ell}, \qquad (3)$$

*where $\binom{K-k}{\ell}$ denotes the binomial coefficient.*

As stated in Definition 3, the configuration of Berlekamp's volume has two hyper-parameters: the total number of queries $K$, and the maximum number of erroneous feedback $K_E$ within all queries. This error mode refers to Berlekamp (1968)'s noisy-channel model. Berlekamp's volume is a tool for designing robust query strategy to guarantee the tolerance of bounded number of feedback errors.

**What type of uncertainty does Berlekamp's volume characterize?** Berlekamp's volume measures the uncertainty of the unknown latent task variable $z$ together with the randomness carried by the noisy oracle feedback. To give a more concrete view, we first show how the value of Berlekamp's volume $\mathcal{BVol}_{\widehat{Z}}(\mathcal{Q}_k)$ changes when receiving the feedback of a new preference query. Depending on the binary oracle feedback for the query trajectory pair $(\tau_k^{(1)}, \tau_k^{(2)})$, the query context set may be updated to two possible statuses:

$$\mathcal{Q}_k^{(1) \succ (2)} = \mathcal{Q}_{k-1} \cup \left\{ (\tau_k^{(1)} \succ \tau_k^{(2)}) \right\} \quad \text{and} \quad \mathcal{Q}_k^{(2) \succ (1)} = \mathcal{Q}_{k-1} \cup \left\{ (\tau_k^{(2)} \succ \tau_k^{(1)}) \right\}, \qquad (4)$$

where $\mathcal{Q}_k^{(u) \succ (v)}$ denotes the updated status in the case of receiving oracle feedback $\tau_k^{(u)} \succ \tau_k^{(v)}$. The relation between $\mathcal{BVol}_{\widehat{Z}}(\mathcal{Q}_{k-1})$ and $\mathcal{BVol}_{\widehat{Z}}(\mathcal{Q}_k)$ is characterized by Proposition 1, which is the foundation of Berlekamp's volume for developing error-tolerant algorithms.

**Proposition 1** (Volume Conservation Law). *For any query context set $\mathcal{Q}_{k-1}$ and arbitrary query trajectory pair $(\tau_k^{(1)}, \tau_k^{(2)})$, the relation between $\mathcal{BVol}_{\widehat{Z}}(\mathcal{Q}_{k-1})$ and $\mathcal{BVol}_{\widehat{Z}}(\mathcal{Q}_k)$ satisfies*

$$\mathcal{BVol}_{\widehat{Z}}(\mathcal{Q}_{k-1}) = \mathcal{BVol}_{\widehat{Z}}(\mathcal{Q}_k^{(1) \succ (2)}) + \mathcal{BVol}_{\widehat{Z}}(\mathcal{Q}_k^{(2) \succ (1)}). \qquad (5)$$

The proofs of all statements presented in this section are deferred to Appendix A. As shown by Proposition 1, each preference query would partition the volume $\mathcal{BVol}_{\widehat{Z}}(\mathcal{Q}_{k-1})$ into two subsequent branches, $\mathcal{BVol}_{\widehat{Z}}(\mathcal{Q}_k^{(1) \succ (2)})$ and $\mathcal{BVol}_{\widehat{Z}}(\mathcal{Q}_k^{(2) \succ (1)})$. The selection of preference queries does not alter the volume sum of subsequent branches. Since the values of $\mathcal{BVol}_{\widehat{Z}}(\cdot)$ are non-negative integers, the volume is monotonically decreased with the online query procedure. When the volume is eliminated to the unit value, the selection of task embedding with minimum mismatches $\mathcal{E}(z; \mathcal{Q}_k)$ would become deterministic (see Proposition 2).

**Proposition 2** (Unit of Volume). *Given a query context set $\mathcal{Q}_k$ with $\mathcal{BVol}_{\widehat{Z}}(\mathcal{Q}_k) = 1$, there exists exactly one task embedding candidate $z \in \widehat{Z}$ satisfying $\mathcal{E}(z; \mathcal{Q}_k) \leq K_E$.*

We can represent the preference-based task inference protocol by a *decision tree*, where each $\mathcal{Q}_k$ with $\mathcal{BV}ol_{\widehat{Z}}(\mathcal{Q}_k) = 1$ corresponds to a *leaf node* and each preference query is a *decision rule*. The value of Berlekamp's volume $\mathcal{BV}ol_{\widehat{Z}}(\mathcal{Q}_k)$ corresponds to the number of leaf nodes remaining in the subtree rooted with context set $\mathcal{Q}_k$. The value of $z$-conditioned volume $vol_z(\mathcal{Q}_k)$ counts the number of leaf nodes with $z$ as the final decision. Each path from an ancestor node to a leaf node can be mapped to a valid feedback sequence that does not violate $\mathcal{E}(z; \mathcal{Q}_K) \leq K_E$. From this perspective, Berlekamp's volume quantifies the uncertainty of task inference by counting the number of valid feedback sequences for the incoming queries.

**Error-Tolerant Query Strategy.** Given Berlekamp's volume as the uncertainty quantification, the query strategy can be constructed directly by maximizing the worst-case uncertainty reduction:

$$\textsc{GenerateQuery}(\widehat{Z}, \mathcal{Q}_{k-1}) = \underset{\tau_k^{(1)}, \tau_k^{(2)}}{\arg\min} \left( \max\left\{ \mathcal{BV}ol_{\widehat{Z}}(\mathcal{Q}_k^{(1) \succ (2)}), \ \mathcal{BV}ol_{\widehat{Z}}(\mathcal{Q}_k^{(2) \succ (1)}) \right\} \right), \quad (6)$$

where $\textsc{GenerateQuery}(\cdot)$ refers to the query generation step at line 5 of Algorithm 1. This design follows the principle of binary search. If the Berlekamp's volumes of two subsequent branches are well balanced, the uncertainty can be exponentially reduced no matter which feedback the oracle responds. In our implementation, the $\arg\min$ operator in Eq. (6) is approximated by sampling a mini-batch of trajectory pairs from the experience buffer. The query content is determined by finding the best trajectory pair within the sampled mini-batch. More implementation details are included in Appendix B.2.

# 4 Experiments

In this section, we investigate the empirical performance of ANOLE on a suite of Meta-RL benchmark tasks. We compare our method with simple preference-based adaptation strategies and conduct several ablation studies to demonstrate the effectiveness of our algorithmic designs.

## 4.1 Experiment Setup

**Experiment Setting.** We adopt six meta-RL benchmark tasks created by Rothfuss et al. (2019), which are widely used by meta-RL works to evaluate the performance of few-shot policy adaptation (Rakelly et al., 2019; Zintgraf et al., 2020; Fakoor et al., 2020). These environment settings consider four ways to vary the task specification $\mathcal{T}$: forward/backward (-Fwd-Back), random target velocity (-Rand-Vel), random target direction (-Rand-Dir), and random target location (-Rand-Goal). We simulate the preference oracle $\Omega_{\mathcal{T}}$ by comparing the ground-truth trajectory return given by the MuJoCo-based environment simulator. The adaptation protocol cannot observe these environmental rewards during meta-testing and can only query the preference oracle $\Omega_{\mathcal{T}}$ to extract the task information. In addition, we impose a random noisy perturbation on the oracle feedback. Each binary feedback would be flipped with probability $\epsilon$. We consider such independently distributed errors rather than the bounded-number error mode to evaluate the empirical performance, since it is more realistic to simulate human's unintended errors. A detailed description of the experiment setting is included in Appendix B.1.

**Implementation of ANOLE.** Note that ANOLE is an adaptation module and can be built upon any meta-RL or multi-task RL algorithms with latent policy encoding (Hausman et al., 2018; Eysenbach et al., 2019; Pong et al., 2020; Lynch et al., 2020; Gupta et al., 2020). To align with the baseline algorithms, we implement a meta-training procedure similar to PEARL (Rakelly et al., 2019). The same as PEARL, our policy optimization module extends soft actor-critic (SAC; Haarnoja et al., 2018) with the latent task embedding $z$, *i.e.*, the policy and value functions are represented by $\pi_\theta(a \mid s; z)$ and $Q_\phi(s, a; z)$ where $\theta$ and $\phi$ denote the parameters. One difference from PEARL is the removal of the inference network, since it is no longer used in meta-testing. Instead, we set up a bunch of trainable latent task variables $\{z_i\}_{i=1}^N$ to learn the multi-task policy. More implementation details are included in Appendix B.2. The source code of our ANOLE implementation and experiment scripts are available at `https://github.com/Stilwell-Git/Adaptation-with-Noisy-OracLE`.

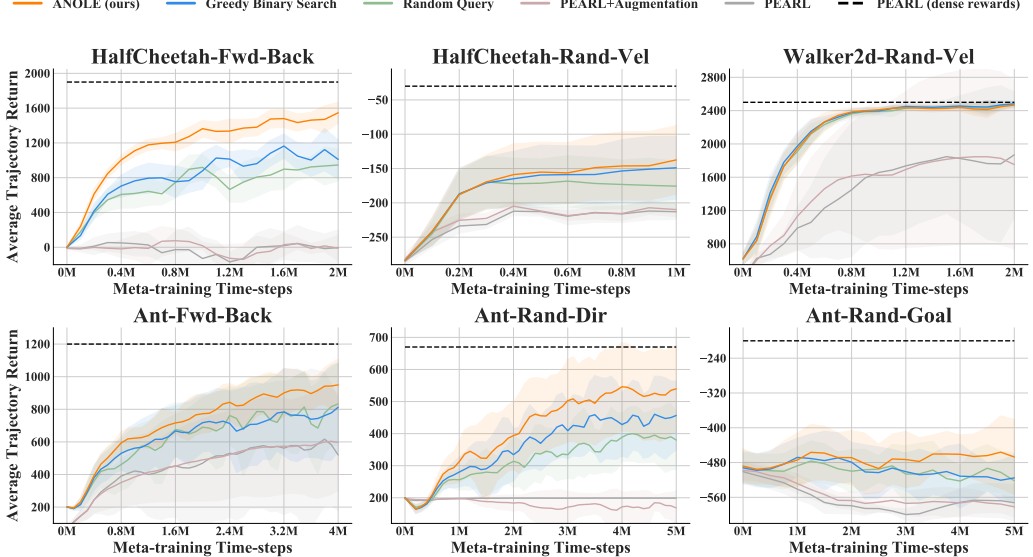

Figure 1: Learning curves on a suite of MuJoCo-based meta-RL benchmark tasks with preference-based adaptation. All curves plot the average performance from eight runs with random initialization. The shaded region indicates the standard deviation. "PEARL (dense rewards)" denotes the final performance of ordinary PEARL using dense reward signals for meta-testing.

**Baseline Algorithms.** We compare the performance with four baseline adaptation strategies. Two of these baselines are built upon the same meta-training pre-processing as ANOLE but do not use Berlekamp's volume to construct preference queries.

- **Greedy Binary Search.** We conduct a simple and direct implementation of binary-search-like task inference. When generating preference queries, it simply ignores all candidates that have made at least one wrong preference predictions, and the query trajectory pair only aims to partition the remaining candidate pool into two balanced separations.

- **Random Query.** We also include the simplest baseline that constructs the preference query by drawing a random trajectory pair from the experience buffer. This baseline serves an ablation study to investigate the benefits of removing the inference network.

In addition, we implement two variants of PEARL (Rakelly et al., 2019) that use a probabilistic context encoder to infer task specification from preference-based feedback.

- **PEARL.** We modify PEARL's context encoder to handle the preference-based feedback. More specifically, the context encoder is a LSTM-based network that takes an ordered trajectory pair as inputs to make up the task embedding $z$, *i.e.*, the input trajectory pair has been sorted by oracle preference. In meta-training, the preference is labeled by the ground-truth rewards from the environment simulator. During adaptation, the PEARL-based agent draws random trajectory pairs from the experience buffer to query the preference oracle, and the oracle feedback is used to update the posterior of task variable $z$.

- **PEARL+Augmentation.** We implement a data augmentation method for PEARL's meta-training to pursue the error tolerance of preference-based inference network. We impose random errors to the preference comparison when training the preference-based context encoder so that the inference network is expected to have some extent of error tolerance.

We include more implementation details of these baselines in Appendix B.3.

## 4.2 Performance Evaluation on MuJoCo-based Meta-RL Benchmark Tasks

Figure 1 presents the performance evaluation of ANOLE and baseline algorithms on a suite of meta-RL benchmark tasks with noisy preference oracle. The adaptation algorithms are restricted to use $K = 10$ preference queries, and the noisy oracle would give wrong feedback for each query

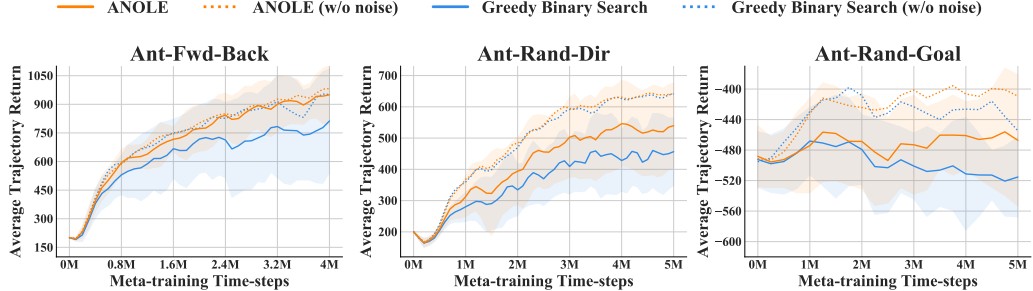

Figure 2: Investigating the impact of oracle noises on performance of preference-based adaptation. The dotted curves with tag "(w/o noise)" refer to the performance evaluation with a noiseless oracle.

with probability $\epsilon = 0.2$. We configure ANOLE with $K_E = 2$ to compute Berlekamp's volume. The experiment results indicate two conclusions:

1. Berlekamp's volume improves the error tolerance of preference-based task inference. The only difference between ANOLE and the first two baselines, *Greedy Binary Search* and *Random Query*, is the utilization of Berlekamp's volume, which leads to a significant performance gap on benchmark tasks.

2. The non-parametric inference framework of ANOLE improves the scalability of preference-based few-shot adaptation. Note that, using ANOLE's algorithmic framework (see Algorithm 1), a random query strategy can also outperform PEARL-based baselines. It may because the inference network used by PEARL cannot effectively translate the binary preference feedback to the task information.

## 4.3 Ablation Studies on the Magnitude of Oracle Noise

To demonstrate the functionality of Berlekamp's volume on improving error tolerance, we conduct an ablation study on the magnitude of oracle noise. In Figure 2, we contrast the performance of ANOLE in adaptation settings with and without oracle noises, *i.e.*, $\epsilon = 0.2$ *vs.* $\epsilon = 0.0$. When the preference oracle does not carry any error, the simple greedy baseline can achieve the same performance as ANOLE. If we impose noises to the oracle feedback, ANOLE would suffer from a performance drop, where the drop magnitude is much more moderate than that of the greedy strategy. This result indicates that Berlekamp's volume does provide remarkable benefits on improving error tolerance whereas cannot completely eliminate the negative effects of oracle noises. It opens up a problem for future work to further improve the error tolerance of preference-based few-shot adaptation. In Appendix C, we conduct more ablation studies to understand the algorithmic functionality of ANOLE.

## 4.4 Experiments with Human Feedback

We conduct experiments with real human feedback on MuJoCo-based meta-RL benchmark tasks. The meta-testing task specifications are generated by the same rule as the experiments in section 4.2. To facilitate human participation, we project the agent trajectory and the goal direction vector to a 2D coordinate system. The human participant watches the query trajectory pair and labels the preference according to the assigned task goal vector. The performance of

Table 1: Evaluating the performance of ANOLE using human feedback.

| Task | ANOLE |
|------|-------|
| HalfCheetah-Fwd-Back | 1734.3±10.3 |
| Ant-Fwd-Back | 931.1±38.0 |
| Ant-Rand-Dir | 644.8±63.2 |

ANOLE with human feedback is presented in Table 1. Each entry is evaluated by 20 runs (*i.e.*, using $20 \times 10 = 200$ human queries). We note that, in these experiments, the feedback accuracy of the human participant is better than the uniform-noise oracle considered in section 4.2. The average error rate of human feedback is 6.2% over all experiments. The evaluation results are thus better than the experiments in Figure 1. We include a visualization of the human-agent interaction interface in Appendix G.

# 5 Related Work

**Meta-Learning and Meta-RL.** Modeling task specification by a latent task embedding (or latent task variable) is a widely-applied technique in meta-learning for both supervised learning (Rusu et al., 2019; Gordon et al., 2019) and reinforcement learning (Rakelly et al., 2019). This paper studies the adaptation of latent task embedding based on preference-based supervision. One characteristic of our proposed approach is the non-parametric nature of the task inference protocol. Non-parametric adaptation has been studied in supervised meta-learning (Vinyals et al., 2016; Snell et al., 2017; Allen et al., 2019) but is rarely applied to meta-RL. From this perspective, our algorithmic framework opens up a new methodology for few-shot policy adaptation.

**Preference-based RL and Human-in-the-loop Learning.** Learning from human's preference ranking is a classical problem formulation of human-in-the-loop RL (Akrour et al., 2011, 2012; Fürnkranz et al., 2012; Wilson et al., 2012). When combining with deep RL techniques, recent advances focus on learning an auxiliary reward function that decomposes the trajectory-wise preference feedback to per-step supervision (Christiano et al., 2017; Ibarz et al., 2018). Several methods have been developed to generate informative queries (Lee et al., 2021b), improve the efficiency of data utilization (Park et al., 2022), and develop specialized exploration for preference-based RL (Liang et al., 2022). In this paper, we open up a new problem for preference-based RL, *i.e.*, preference-based few-shot policy adaptation. The potential applications of our problem setting may be similar to that of personalized adaptation (Yu et al., 2021; Wang et al., 2021), a supervised meta-learning problem for modeling user preference. A future work is considering a wider range of human supervision, such as human attention (Zhang et al., 2020) and human annotation (Guan et al., 2021).

# 6 Conclusion and Discussions

In this paper, we study the problem of few-shot policy adaptation with preference-based feedback and propose a novel meta-RL algorithm, called *Adaptation with Noisy OracLE* (ANOLE). Our method leverages a classical problem formulation called Rényi-Ulam's game to model the task inference problem with a noisy preference oracle. This connection to information theory enable us to extend the technique of Berlekamp's volume to establish an error-tolerant approach for preference-based task inference, which is demonstrated as a promising approach on an extensive set of benchmark tasks.

We conclude this paper by discussing limitations, future works, and other relevant aspects that have not been covered.

**Adaptation to Environment Dynamics.** One limitation of ANOLE is that it does not consider the potential shift of transition dynamics from meta-training environments to meta-testing environments. Our problem formulation assumes only the reward function alters in the adaptation phase (see section 2.1). The current implementation of ANOLE cannot handle the adaptation to the changing environment dynamics. One promising way to address this issue is to integrate ANOLE with classical meta-RL modules to model the probabilistic inference regarding the transition function. In addition, it is critical to investigate the behavior of human preference when the query trajectories may contain unrealistic transitions.

**Expressiveness of Preference Partial Ordering.** A recent theoretical work indicates that the trajectory-wise partial-order preference can define richer agent behaviors than step-wise reward functions (Abel et al., 2021). However, this superior of preference-based learning has rarely been shown in the empirical studies. *e.g.*, in our experiments, the preference feedback is simulated by summing step-wise rewards given by the MuJoCo simulator, which is a common setting used by most preference-based RL works. In addition, most advanced preference-based RL algorithms are built on a methodology that linearly decomposes the trajectory-wise preference supervision to a step-wise auxiliary reward function, which degrades the expressiveness of the partial ordering preference system. These problems are fundamental challenges to preference-based learning.

## Acknowledgments and Disclosure of Funding

This work is supported by National Key R&D Program of China No. 2021YFF1201600.

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
