# A Omitted Proofs

The proofs of these propositions are extended from Berlekamp (1968).

**Proposition 1** (Volume Conservation Law). *For any query context set $\mathcal{Q}_{k-1}$ and arbitrary query trajectory pair $(\tau_k^{(1)}, \tau_k^{(2)})$, the relation between $\mathcal{BV}ol_{\widehat{Z}}(\mathcal{Q}_{k-1})$ and $\mathcal{BV}ol_{\widehat{Z}}(\mathcal{Q}_k)$ satisfies*

$$\mathcal{BV}ol_{\widehat{Z}}(\mathcal{Q}_{k-1}) = \mathcal{BV}ol_{\widehat{Z}}(\mathcal{Q}_k^{(1)\succ(2)}) + \mathcal{BV}ol_{\widehat{Z}}(\mathcal{Q}_k^{(2)\succ(1)}). \tag{5}$$

*Proof.* Note that both oracle's preference feedback and $z$-conditioned preference prediction are binary values. *i.e.*, for a given preference query $(\tau_k^{(1)}, \tau_k^{(2)})$, the prediction given by each task embedding $z$ is either correct or wrong. The mismatch count must be updated to one of the following cases:

- Case #1: $\mathcal{E}(z; \mathcal{Q}_k^{(1)\succ(2)}) = \mathcal{E}(z; \mathcal{Q}_{k-1})$ and $\mathcal{E}(z; \mathcal{Q}_k^{(2)\succ(1)}) = \mathcal{E}(z; \mathcal{Q}_{k-1}) + 1$;

- Case #2: $\mathcal{E}(z; \mathcal{Q}_k^{(2)\succ(1)}) = \mathcal{E}(z; \mathcal{Q}_{k-1})$ and $\mathcal{E}(z; \mathcal{Q}_k^{(1)\succ(2)}) = \mathcal{E}(z; \mathcal{Q}_{k-1}) + 1$.

It implies that

$$
\begin{aligned}
vol_z(\mathcal{Q}_{k-1}) &= \sum_{\ell=0}^{K_E - \mathcal{E}(z; \mathcal{Q}_{k-1})} \binom{K - (k-1)}{\ell} \\
&= \sum_{\ell=0}^{K_E - \mathcal{E}(z; \mathcal{Q}_{k-1})} \left( \binom{K-k}{\ell} + \binom{K-k}{\ell-1} \right) \\
&= \sum_{\ell=0}^{K_E - \mathcal{E}(z; \mathcal{Q}_{k-1})} \binom{K-k}{\ell} + \sum_{\ell=0}^{K_E - (\mathcal{E}(z; \mathcal{Q}_{k-1})+1)} \binom{K-k}{\ell} \\
&= \sum_{\ell=0}^{K_E - \mathcal{E}(z; \mathcal{Q}_k^{(1)\succ(2)})} \binom{K-k}{\ell} + \sum_{\ell=0}^{K_E - \mathcal{E}(z; \mathcal{Q}_k^{(2)\succ(1)})} \binom{K-k}{\ell} \\
&= vol_z(\mathcal{Q}_k^{(1)\succ(2)}) + vol_z(\mathcal{Q}_k^{(2)\succ(1)}).
\end{aligned}
$$

By plugging into Eq. (3), we have

$$
\begin{aligned}
\mathcal{BV}ol_{\widehat{Z}}(\mathcal{Q}_{k-1}) &= \sum_{z \in \widehat{Z}} vol_z(\mathcal{Q}_{k-1}) \\
&= \sum_{z \in \widehat{Z}} \left( vol_z(\mathcal{Q}_k^{(1)\succ(2)}) + vol_z(\mathcal{Q}_k^{(2)\succ(1)}) \right) \\
&= \mathcal{BV}ol_{\widehat{Z}}(\mathcal{Q}_k^{(1)\succ(2)}) + \mathcal{BV}ol_{\widehat{Z}}(\mathcal{Q}_k^{(2)\succ(1)}).
\end{aligned}
$$

$\square$

**Proposition 2** (Unit of Volume). *Given a query context set $\mathcal{Q}_k$ with $\mathcal{BV}ol_{\widehat{Z}}(\mathcal{Q}_k) = 1$, there exists exactly one task embedding candidate $z \in \widehat{Z}$ satisfying $\mathcal{E}(z; \mathcal{Q}_k) \leq K_E$.*

*Proof.* Note that the values of $\mathcal{BV}ol_{\widehat{Z}}(\cdot)$ and $vol_z(\cdot)$ are non-negative integers. $\mathcal{BV}ol_{\widehat{Z}}(\mathcal{Q}_k) = 1$ implies there exists exactly one task embedding $z \in \widehat{Z}$ satisfying $vol_z(\mathcal{Q}_k) = 1 = \binom{0}{0}$, which further implies $\mathcal{E}(z; \mathcal{Q}_k) = K_E$. $\square$

# B  Experiment Setting and Implementation Details

## B.1  Experiment Setting

We adopt the environment setting created by Rothfuss et al. (2019). This benchmark is a suite of MuJoCo locomotion tasks, where the reward function are varied to create a multi-task setting. More specifically, there are four ways to vary the task specification $\mathcal{T}$:

- `Fwd-Back`: The task variable $\mathcal{T}$ varies the target direction within {forward/backward};
- `Rand-Vel`: The task variable $\mathcal{T}$ varies the target velocity within a bounded range;
- `Rand-Dir`: The task variable $\mathcal{T}$ varies the target direction within the 2D-plane;
- `Rand-Goal`: The task variable $\mathcal{T}$ varies the target location within a bounded area.

The training and testing tasks are randomly generated by a fixed random seed. *i.e.*, the generation of training/testing tasks do not vary across runs. During meta-training, the meta-RL algorithm has the full access to the environmental interaction. The algorithms can obtain trajectories with both transition and reward information. During meta-testing, the reward function would become unavailable to the meta-RL agent. The agent can only query a preference oracle to extract information about the task specification $\mathcal{T}$. The preference oracle $\Omega_{\mathcal{T}}$ is simulated by comparing the ground-truth trajectory return given by the MuJoCo simulator, *i.e.*, the oracle can access to the ground-truth reward function. We consider this asymmetric supervision setting since this paper only focus on the design of adaptation protocol, and our proposed adaptation algorithm can be plugged in any meta-training algorithm using latent embeddings.

In our experiments, all networks are trained using a single GPU and a single CPU core.

- GPU: GeForce GTX 1080 Ti;
- CPU: Intel(R) Xeon(R) CPU E5-2630 v4 @ 2.20GHz.

In each run of experiment, 4M steps of training can be completed within 24 hours.

## B.2  Implementation Details of ANOLE

The overall meta-training procedure of ANOLE is implemented upon PEARL (Rakelly et al., 2019) with some modifications and incremental designs.

**Probabilistic Embedding.**   Note that, different from most PEARL-based algorithms, ANOLE does not include an inference network. Instead, we use a set of trainable variables to model the latent task embedding of each training task. To expand the latent space and promote generalization, we assign each training task $\mathcal{T}_i$ a multivariate Gaussian $\mathcal{N}(\mu_i, \sigma_i)$ with zero covariance, where $\mu_i, \sigma_i^2 \in \mathbb{R}^d$ and $d$ denotes the dimension of latent space. More specifically, we set up $2dN$ trainable variables instead of an inference network to model the latent space from training tasks $\{\mathcal{T}_i\}_{i=1}^N$. The same as PEARL, we conduct a regularization loss to make the learned latent space compact:

$$\mathcal{L}^{\text{KL}}(\mu, \sigma) = \frac{1}{N} \sum_{i=1}^{N} D_{\text{KL}}(\mathcal{N}(\mu_i, \sigma_i^2) \,\|\, \mathcal{N}(\mathbf{0}_d, \mathbf{1}_d)),$$

where $\mathcal{N}(\mathbf{0}_d, \mathbf{1}_d)$ denotes a $d$-dimensional standard multivariate Gaussian.

**Policy Training.**   We adopt the same off-policy meta-RL framework as PEARL to train the policy. We extend soft actor-critic (SAC; Haarnoja et al., 2018) with the latent task embedding $z$, *i.e.*, the policy and value functions are represented by $\pi_\theta(a \mid s; z)$ and $Q_\phi(s, a; z)$ where $\theta$ and $\phi$ denote the parameters. The objective function for actor and critic networks are presented as below:

$$\mathcal{L}^{\text{actor}}(\theta) = \mathop{\mathbb{E}}_{(\mathcal{T}_i, s) \sim \mathcal{D}} \left[ D_{\text{KL}} \left( \pi_\theta(\cdot|s) \,\Big\|\, \frac{\exp(Q_\phi(s, \cdot))}{\int \exp(Q_\phi(s, a)) \mathrm{d}a} \right) \Big| z \sim \mathcal{N}(\mu_i, \sigma_i^2) \right],$$

$$\mathcal{L}^{\text{critic}}(\phi) = \mathop{\mathbb{E}}_{(\mathcal{T}_i, s, a, r, s') \sim \mathcal{D}} \left[ \left( Q_\phi(s, a; z) - r - \gamma V_{\phi_{\text{target}}}(s'; z) \right)^2 \Big| z \sim \mathcal{N}(\mu_i, \sigma_i^2) \right].$$

This part of implementation, including network architecture and optimizers, is reused from the open-source code of PEARL.

**Preference Predictor.** We train a $z$-conditioned preference predictor $f_\psi(\cdot; z)$ on the meta-training tasks by optimizing the preference loss function:

$$\mathcal{L}^{\text{Pref}}(\psi) = \mathbb{E}\left[ D_{\text{KL}}\left( \mathbb{I}\left[\tau^{(1)} \succ \tau^{(2)} \mid \mathcal{T}\right] \,\Big\|\, f_\psi\left(\tau^{(1)} \succ \tau^{(2)}; z\right) \right) \right],$$

where $\psi$ denotes the parameters of the preference predictor, $D_{\text{KL}}(\cdot\|\cdot)$ denotes the Kullback–Leibler divergence, and $\mathbb{I}[\tau^{(1)} \succ \tau^{(2)} \mid \mathcal{T}]$ is the ground-truth preference order specified by the task specification $\mathcal{T}$ (*e.g.*, specified by the reward function on training tasks). The trajectory pair $\langle \tau^{(1)}, \tau^{(2)} \rangle$ is drawn from the experience buffer, and $z$ is the task embedding vector encoding $\mathcal{T}$. Optimizing this KL-based loss function is equivalent to optimizing binary cross entropy.

Following the implementation of preference-based RL (Christiano et al., 2017; Lee et al., 2021b), we use Bradley-Terry model (Bradley and Terry, 1952) to establish a preference predictor:

$$f_\psi(\tau^{(1)} \succ \tau^{(2)}; z) = \frac{\exp\left(\sum_t g_\psi(s_t^{(1)}, a_t^{(1)}; z)\right)}{\exp\left(\sum_t g_\psi(s_t^{(1)}, a_t^{(1)}; z)\right) + \exp\left(\sum_t g_\psi(s_t^{(2)}, a_t^{(2)}; z)\right)}, \tag{7}$$

where $g_\psi(s, a; z)$ is a network that outputs the ranking score of state-action pair $(s, a)$. In implementation, $\tau^{(1)}$ and $\tau^{(2)}$ refer to two fixed-length trajectory segments instead of considering the complete trajectory. A future work is adopting the random-sampling trick (Ren et al., 2022) for the extension to long-horizon preference.

**Batch-Constrained Embedding Sampling.** A pre-processing step of our task inference algorithm is sampling a candidate pool of task embeddings (see line 2 in Algorithm 1) for the subsequent embedding selection. We restrict the support of this candidate pool to the task embedding distribution conducted in meta-training. We call this procedure batch-constrained embedding sampling, since it corresponds to the notion of batch-constrained policy (Fujimoto et al., 2019) in the literature of offline reinforcement learning. This restriction ensures the induced policy $\pi_\theta(a|s; z)$ is covered by the training distribution, so that the meta-testing policies would not suffer from unpredictable out-of-distribution generalization errors. More specifically, we sample a set of task embeddings from the mixture distribution of training tasks, $\widehat{Z} \leftarrow \{z_j \sim q(z)\}_{j=1}^M$ where $q(z)$ refers to the mixture of training task variable:

$$q(z) = \frac{1}{N}\sum_{i=1}^N \mathcal{N}(z \mid \mu_i, \sigma_i^2).$$

**Candidate Pool Size.** Note that the number of embedding candidates initialized in $\widehat{Z}$ may affect the computation of Berlekamp's volume. The size of candidate pool $\widehat{Z}$, denoted by $M$, is determined by the following formula:

$$M = \left\lfloor \frac{2^K}{vol_z(\emptyset)} \right\rfloor = \left\lfloor \frac{2^K}{\sum_{\ell=0}^{K_E} \binom{K}{\ell}} \right\rfloor,$$

where $z$ is an arbitrary embedding candidate. This configuration of candidate pool size ensures that, with ideal preference queries, Berlekamp's volume $\mathcal{BVol}_{\widehat{Z}}(\mathcal{Q}_K)$ can be reduced to 1 by $K$ queries. *i.e.*, in the ideal situation, each preference query can halve the value of $\mathcal{BVol}_{\widehat{Z}}(\mathcal{Q}_k)$. Note that, our algorithm does not require $\mathcal{BVol}_{\widehat{Z}}(\mathcal{Q}_K)$ to be reduced to 1, since finding the task embedding with minimum mismatch is always a plausible solution (see $\arg\min_{z \in \widehat{Z}} \mathcal{E}(z; \mathcal{Q}_K)$ at line 8 of Algorithm 1).

**Error-Tolerant Query Strategy.** In our implementation, we use mini-batch sampling to approximate the query generation protocol GENERATEQUERY($\cdot$) in Eq. (6). We sample a mini-batch of trajectory pairs and find the best trajectory pair within the sampled mini-batch.

$$\text{GENERATEQUERY}(\widehat{Z}, \mathcal{Q}_{k-1}) = \underset{(\tau_k^{(1)}, \tau_k^{(2)}) \in B}{\arg\min}\left( \max\left\{ \mathcal{BVol}_{\widehat{Z}}(\mathcal{Q}_k^{(1)\succ(2)}), \ \mathcal{BVol}_{\widehat{Z}}(\mathcal{Q}_k^{(2)\succ(1)}) \right\} \right),$$

where $B$ denotes a mini-batch of trajectory pairs that are uniformly sampled from the experience replay buffer. In our implementation, we sample 100 trajectory pairs for each mini-batch $B$.

**Hyper-Parameters.** We summarize major hyper-parameters as the following table. We use this set of hyper-parameters for all ANOLE's experiments.

| Hyper-Parameter | Default Configuration |
|---|---|
| dimension of latent embedding $d$ | 5 |
| discount factor $\gamma$ | 0.99 |
| optimizer (all losses) | Adam (Kingma and Ba, 2015) |
| learning rate | $3 \cdot 10^{-4}$ |
| Adam-$(\beta_1, \beta_2, \epsilon)$ | $(0.9, 0.999, 10^{-8})$ |
| temperature $\alpha$ | 1.0 |
| Polyak-averaging coefficient | 0.005 |
| # gradient steps per environment step | 1/5 |
| # gradient steps per target update | 1 |
| # transitions in replay buffer (for each task $\mathcal{T}$) | $10^6$ |
| # tasks in each mini-batch for training SAC | 10 |
| # transitions in each task-batch for training SAC | 256 |
| # trajectory segments in each mini-batch for training $f_\psi$ | 10 |
| # transitions in each trajectory segment for training $f_\psi$ | 64 |
| # preference queries $K$ | 10 |
| # wrong feedbacks to tolerate $K_E$ | 2 |
| # trajectory pairs to approximate GENERATEQUERY | 100 |

## B.3 Implementation Details of PEARL-based Baselines

We slightly modify the implementation of PEARL to make it work for preference-based adaptation.

**Preference-based Context Encoder.** We modify PEARL's context encoder to handle the preference-based feedback. We use a LSTM-based context encoder that takes an ordered trajectory pair as inputs to make up the task embedding $z$, *i.e.*, the input trajectory pair has been sorted by oracle preference. The architecture of LSTM-based encoder is implemented by the open-source code of PEARL. The same as the original version of PEARL, the output of context encoder is a $d$-dimensional Gaussian. In meta-training, the preference is labeled by the ground-truth rewards from the environment simulator, *i.e.*, comparing the sum of ground-truth rewards. During adaptation, the PEARL-based agent draws random trajectory pairs from the experience buffer to query the preference oracle, and the oracle feedback is used to update the posterior of task variable $z$. The posterior update rule is reused from the open source code of PEARL.

**Data Augmentation for Error-Tolerance.** We implement a data augmentation method for PEARL's meta-training to pursue the error tolerance of preference-based inference network. To mimic the error mode of noisy preference oracle, we impose random errors to the preference comparison when training the preference-based context encoder. The error is uniformly flipped preference feedback with probability 0.2. In this way, the preference-based context encoder is trained using the same noisy preference signals as the meta-testing procedure so that the inference module is expected to have some extent of error tolerance.

**Hyper-Parameters.** We do not modify any hyper-parameters of PEARL. Note that the open-source implementation of PEARL specializes different hyper-parameter configurations for different environments. In this paper, we conduct two environments, `Walker2d-Rand-Vel` and `And-Rand-Dir`, without official hyper-parameter configuration since PEARL does not evaluate on them. To address this issue, we transfer hyper-parameter configurations from similar tasks. For `Walker2d-Rand-Vel`, we use the same configuration as `HalfCheetah-Rand-Vel`. For `And-Rand-Dir`, we use the same configuration as `And-Rand-Goal`.

# C   Ablation Studies on the Magnitude of Oracle Noise

We evaluate the performance of ANOLE and baselines under different magnitudes of oracle noises $\epsilon$. These results are generated by a same set of runs. *i.e.*, each run of meta-training evaluates several meta-testing configurations. When the oracle carries no error, the performance of ANOLE and greedy binary search are almost the same. With the noise magnitude increasing, the gap between ANOLE and baselines become larger.

## C.1   Performance Evaluation without Noises ($\epsilon = 0.0$)

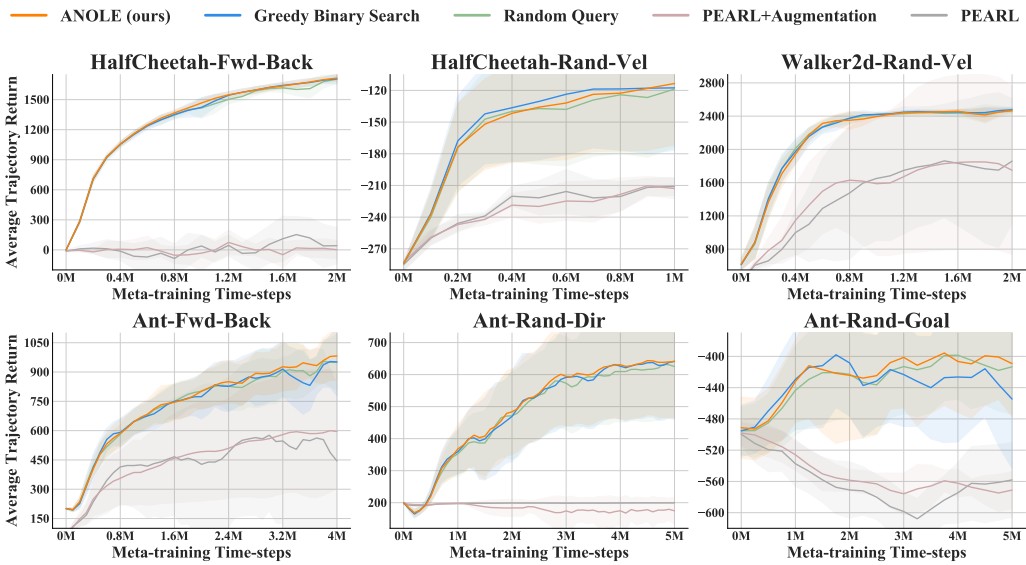

## C.2   Performance Evaluation with Noise Magnitude $\epsilon = 0.1$

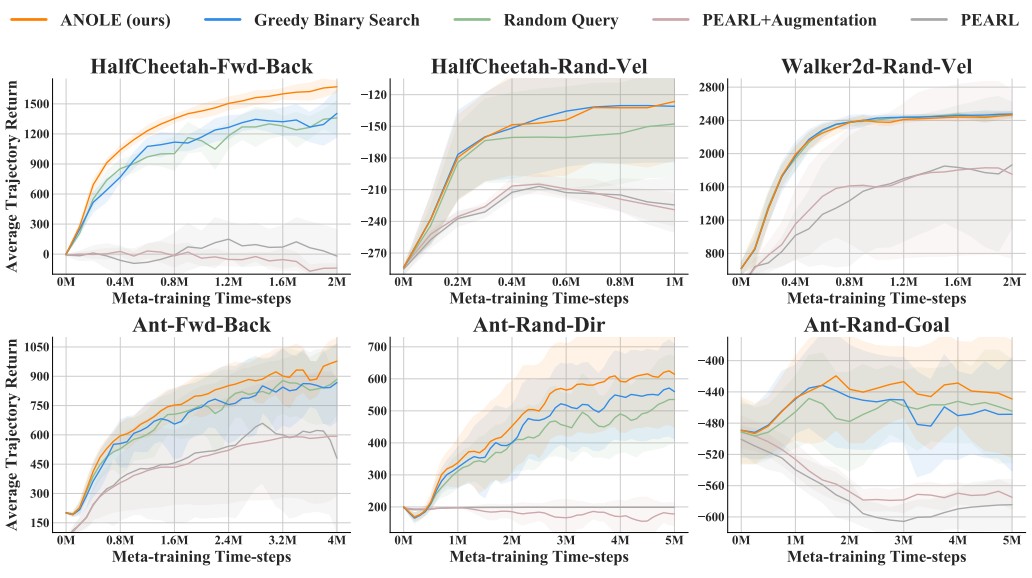

## C.3 Performance Evaluation with Noise Magnitude $\epsilon = 0.2$ (Default Setting)

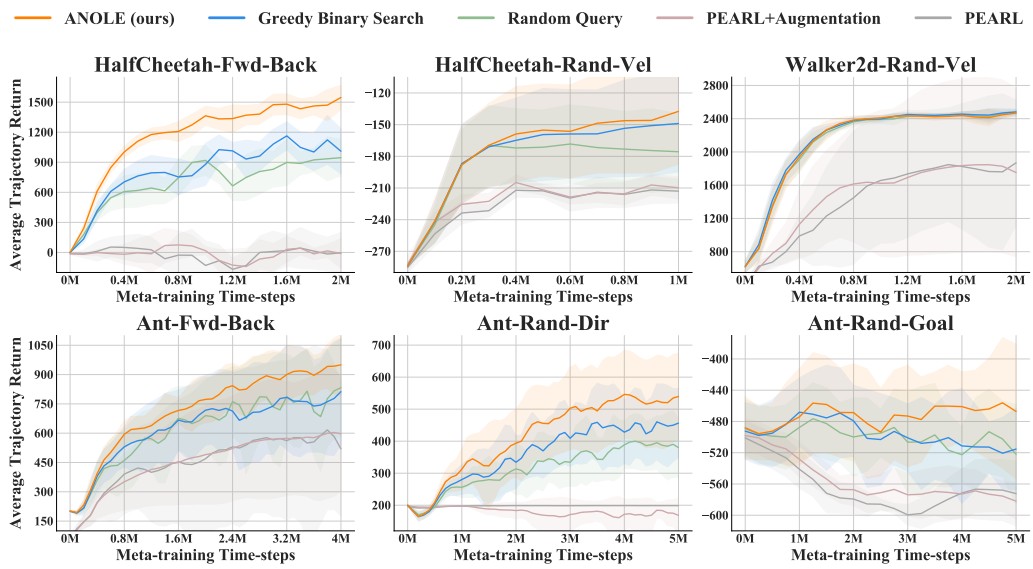

## C.4 Performance Evaluation with Noise Magnitude $\epsilon = 0.3$

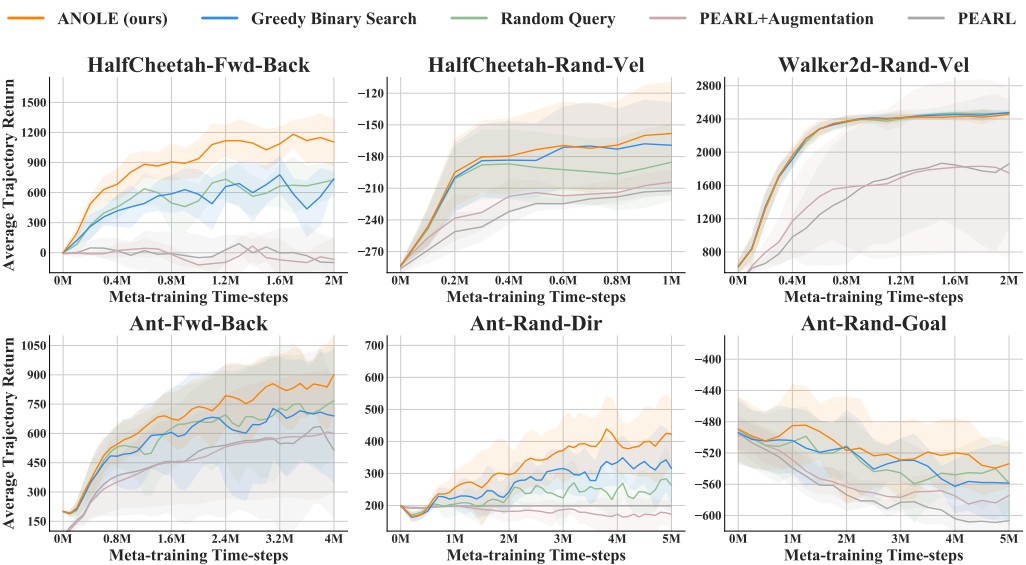

# D   Performance Evaluation at Each Adaptation Step

In addition to the full learning curves, we plot the performance of final policy in every adaptation steps. Final policy refers to the last evaluation point presented in Figure 1. The experiments show that PEARL-based baselines cannot effectively extract task information from preference-based binary feedback.

## D.1   Preference-based Few-shot Adaptation with Noise Magnitude $\epsilon = 0.2$ (Default Setting)

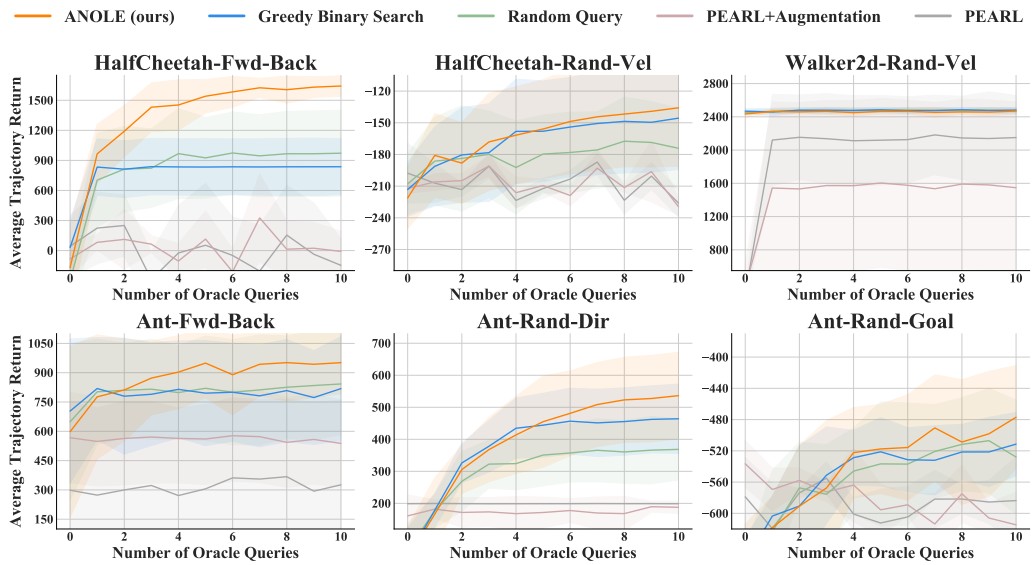

## D.2   Preference-based Few-shot Adaptation without Noises ($\epsilon = 0.0$)

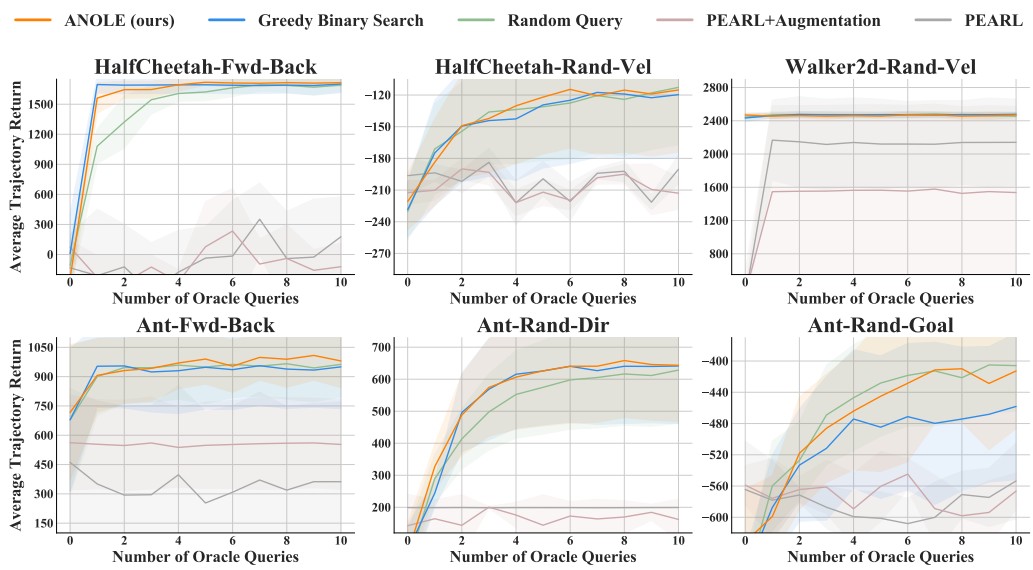

# E  ANOLE with An Alternative Meta-Training Module

We investigate the performance of ANOLE with an alternative meta-training module. We use preference-based supervision to train the meta-policy. Following a classical paradigm of preference-based RL (Christiano et al., 2017), we conduct a reward learning module to decompose the preference-based binary feedback into per-step reward supervision. More specifically, we use the ranking score $g_\psi(s, a; z)$ defined in Eq. (7) as an auxiliary reward function, which is learned from preference comparisons. The experiments show that ANOLE with preference-based meta-training can significantly outperform PEARL-based baselines using environmental rewards.

## E.1  Performance Evaluation with Noise Magnitude $\epsilon = 0.2$ (Default Setting)

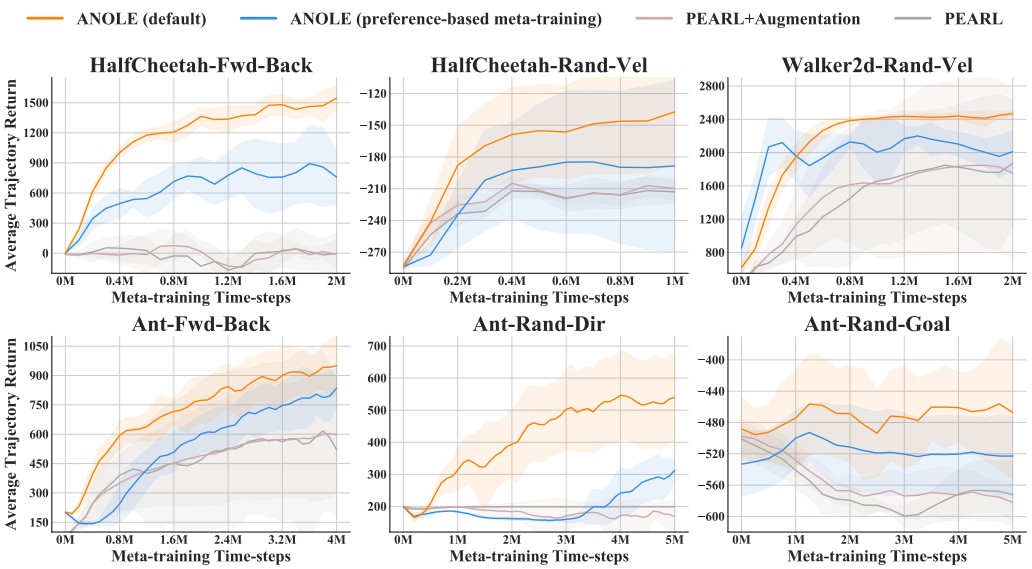

## E.2  Performance Evaluation without Noises ($\epsilon = 0.0$)

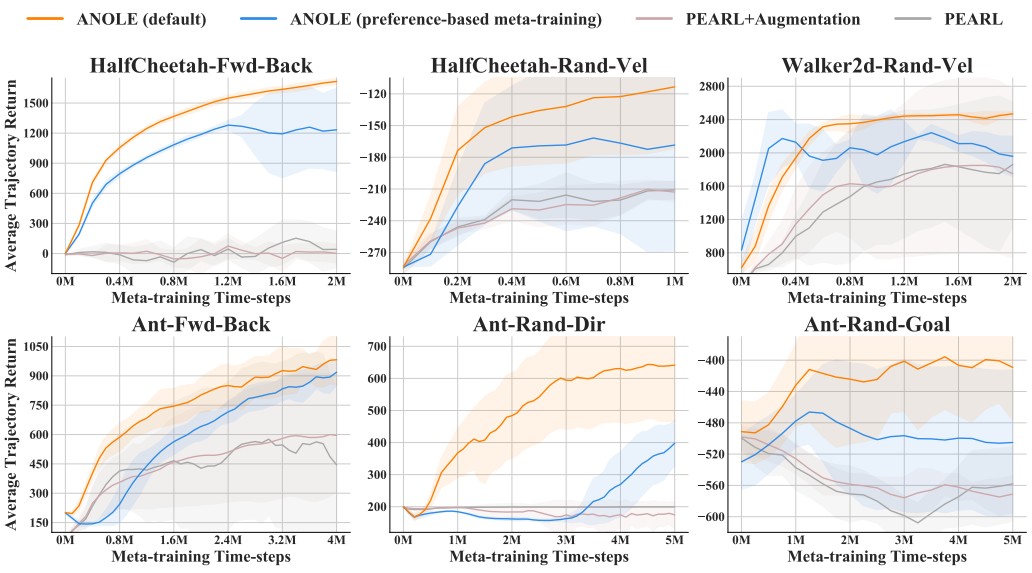

# F   Performance Evaluation under Two Alternative Noise Modes

In addition to the uniform noise mode considered in the main paper, we conduct experiments with two additional noise modes:

- **Boltzman Mode.** The oracle answers $\tau^{(1)} \succ \tau^{(2)}$ with the probability

$$\frac{\exp\left(\beta \cdot \text{Return}(\tau^{(1)})\right)}{\exp\left(\beta \cdot \text{Return}(\tau^{(1)})\right) + \exp\left(\beta \cdot \text{Return}(\tau^{(2)})\right)} = \frac{\exp\left(\beta \sum_t r_t^{(1)}\right)}{\exp\left(\beta \sum_t r_t^{(1)}\right) + \exp\left(\beta \sum_t r_t^{(2)}\right)}$$

  where $\beta$ denotes the temperature parameter. This error mode is commonly considered by recent preference-based RL works (Lee et al., 2021a).
- **Hack Mode.** The oracle always gives wrong feedbacks for the first 20% queries and keeps correct for the remaining 80%. This noise mode is designed to hack search-based query strategies, since the first few queries are usually the most informative.

The experiments results are presented as follows. The learning curves tagged with label $(\beta = \cdot)$ correspond to the experiments with Boltzman noise mode. The learning curves in the last column correspond to the experiments with "Hack" noise mode.

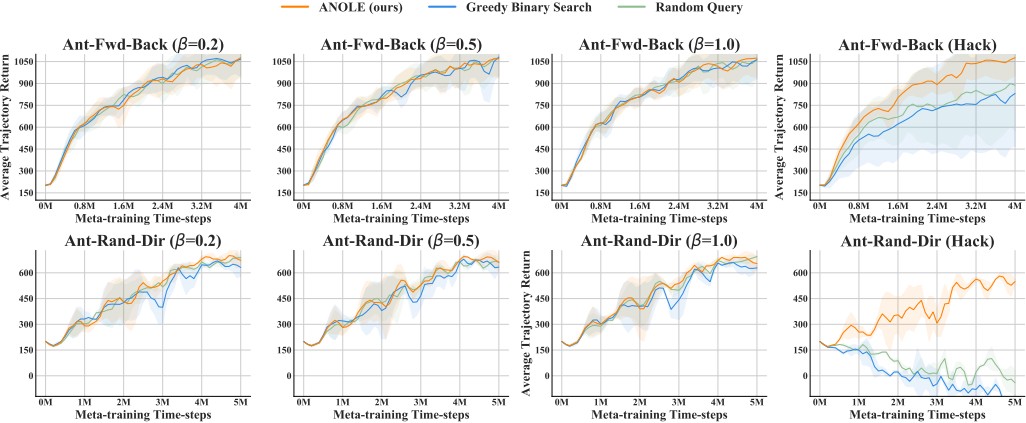

We note that, for these testing environments, Boltzman feedbacks are much more accurate than the uniform-noise oracle considered in the paper. That is because the generated query trajectory pairs can usually be clearly compared. The performance of ANOLE and greedy/random query strategies are comparable since errors hardly occur. Regarding this result, we would like to emphasize that the main purpose of ANOLE is to tolerate unintended errors of human users. More specifically, a robust algorithm is expected to tolerate a few amounts of irrational errors.

# G  Interface of Human-ANOLE Interaction

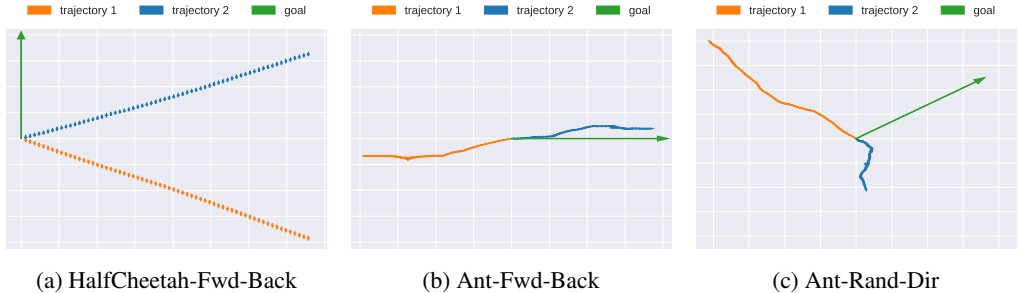

(a) HalfCheetah-Fwd-Back      (b) Ant-Fwd-Back      (c) Ant-Rand-Dir

To facilitate human participation, we project the agent trajectory and the goal direction vector to a 2D coordinate system, *i.e.*, extracting the agent location coordinate from the state representation. The human participant watches the query trajectory pair and labels the preference according to the assigned task goal vector. We implement this interface for three tasks: `HalfCheetah-Fwd-Back`, `Ant-Fwd-Back`, `Ant-Rand-Dir`. Since the `HalfCheetah` agent can only move in a single dimension, we fill in the $x$-axis of `HalfCheetah-Fwd-Back` interface by the timestep index.