# OpenReview forum: "Efficient Meta Reinforcement Learning for Preference-based Fast Adaptation"
_NeurIPS.cc/2022/Conference — NeurIPS 2022 Accept_

### Official Review · Reviewer_L5bT · 2022-07-11

**Rating:** 6
**Confidence:** 4
**Soundness:** 3 good
**Presentation:** 3 good
**Contribution:** 3 good

**Summary:**

This work studies meta-RL with preference-based reward learning at meta-test time. The authors interpret this problem as classical problem formulation called Rényi-Ulam’s game and design a query selection method based on the technique of Berlekamp’s volume. The authors verified the effectiveness of proposed methods on standard RL benchmarks.

**Questions:**

* There are various uncertainty-based sampling for improving preference-based RL. For example, one can utilize the uncertainty from reward ensembles. It would be nice if the authors could include more baselines in the experiments.

* Even though experiments on locomotion tasks are interesting, it would also be nice to include robotics tasks. I think Meta-world (https://github.com/meta-world/meta-world.github.io) can be a good candidate

* Som quantitative results which visualize the selected queries from the proposed method and baseline can be interesting to understand the effects of ANOLE.

* Other noisy feedback. In addition to iid noise (flipping with the probability of epsilon), experiments on other noisy feedback can be interesting. For example, authors can consider the Boltzman preference model.

**Limitations:**

I'd like to recommend including real-human experiments. Because real humans can show various irrationality and have an inductive bias in decision making, the trends on real humans can be different from synthetic preferences. Therefore, it would be nice if the authors could include real human experiments.

**Strengths And Weaknesses:**

# Strength

* Interesting research problem: Meta-RL with preference-based learning is a very interesting research problem, which can be useful for utilizing RL in challenging domains.

* Clear writing: the paper is well-written and easy to follow.

* Interesting connection to Rényi-Ulam’s game & technique of Berlekamp’s volume to generate queries.

# Weakness

* Lack of real human experiments

* Lack of comparison with several baselines

# Overall

The paper is well-written and the proposed method sounds reasonable. However, more experiments are required to make this work more interesting. My recommendation is "boardline accept".

---

> ### Author Response · Authors · 2022-08-02
> **Response to Reviewer L5bT**
>
> Thanks for the comments. We provide clarification to your questions and concerns as below. If our response does not fully address your concerns, please post additional questions and we will be happy to have further discussions.
>
> **Q1: Comparison to the uncertainty quantification given by reward ensembles.**
>
> We conduct experiments using the uncertainty given by reward ensembles. The overall implementation is based on greedy binary search, in which we use the variance of reward model ensembles to weight the task embedding candidates. The experiment result is presented in the following table.
>
> | Hack Mode        | reward ensemble | ANOLE  | Greedy | Random |
> | ---------------- | --------------- | ------ | ------ | ------ |
> | **Ant-Fwd-Back** | 778.06          | 949.74 | 812.52 | 833.16 |
> | **Ant-Rand-Dir** | 419.90          | 539.62 | 456.51 | 379.93 |
>
> The result shows that weighting by reward ensembles would even hurt the performance of greedy binary search. We would like to clarify that the uncertainty of reward model ensembles only characterizes the richness of training data for the corresponding state-action pairs. In comparison, Berlekamp's volume considers the randomness of oracle behavior rather than the randomness of environment, which makes Berlekamp's volume become a special tool and cannot be replaced by simple uncertainty quantification.
>
> **Q2: Experiments on Meta-World manipulation tasks.**
>
> We conduct experiments on two Meta-World manipulation tasks. We present the average return values of ANOLE and baselines in the following table.
>
> | Meta-World Benchmark  | ANOLE | Greedy | Random |
> | --------------------- | ----- | ------ | ------ |
> | **ML1-Push-v2**       | 89.19 | 53.75  | 48.35  |
> | **ML1-Pick-Place-v2** | 8.96  | 8.97   | 8.93   |
>
> These results are recorded when training $10^7$ environment steps. We note that, in the Meta-World benchmark paper [1], PEARL requires $3\times 10^8$ environment steps to achieve a remarkable manipulation success rate in these two tasks. Our computation resource cannot support us to complete the whole training procedure during the short rebuttal period. We will continue to work on it.
>
> [1] Yu, T., Quillen, D., He, Z., Julian, R., Hausman, K., Finn, C., & Levine, S. (2020, May). Meta-world: A benchmark and evaluation for multi-task and meta reinforcement learning. In *Conference on Robot Learning* (pp. 1094-1100). PMLR.
>
> **Q3: Experiments on Boltzman preference model and beyond.**
>
> We conduct experiments on Boltzman preference model, where the Boltzman preference is constructed following the benchmark paper [2]. The oracle answers $\tau_1\succ\tau_2$ with the probability $\frac{\exp(\beta\cdot \text{Return}(\tau_1))}{\exp(\beta\cdot \text{Return}(\tau_1))+\exp(\beta\cdot \text{Return}(\tau_2))}$ where $\beta$ denotes the temperature parameter. The experiment results are presented in the following table.
>
> | Boltzman Mode                  | ANOLE   | Greedy  | Random  |
> | ------------------------------ | ------- | ------- | ------- |
> | **Ant-Fwd-Back** ($\beta=1.0$) | 1090.64 | 1070.13 | 1071.33 |
> | **Ant-Fwd-Back** ($\beta=0.5$) | 1094.65 | 1085.91 | 1050.14 |
> | **Ant-Fwd-Back** ($\beta=0.2$) | 1065.32 | 1044.90 | 1033.38 |
> | **Ant-Rand-Dir** ($\beta=1.0$) | 668.31  | 655.60  | 666.37  |
> | **Ant-Rand-Dir** ($\beta=0.5$) | 644.24  | 605.30  | 606.54  |
> | **Ant-Rand-Dir** ($\beta=0.2$) | 609.20  | 595.39  | 568.62  |
>
> We note that, for these testing environments, Boltzman feedbacks are much more accurate than the uniform-noise oracle considered in the paper. That is because the generated query trajectory pairs can usually be clearly compared. The performance of ANOLE and greedy/random query strategies are comparable since errors hardly occur.
>
> Regarding this result, we would like to emphasize that the main purpose of ANOLE is to tolerate unintended errors of human users. More specifically, a robust algorithm is expected to tolerate a few amounts of irrational errors. To support this point, we design another noisy oracle, called "Hack Mode", which always gives wrong feedbacks for the first 20\% queries and keeps correct for the remaining 80\%. This noise mode is designed to hack search-based query strategies, since the first few queries are usually the most informative.
>
> | Hack Mode        | ANOLE  | Greedy | Random |
> | ---------------- | ------ | ------ | ------ |
> | **Ant-Fwd-Back** | 945.41 | 433.52 | 652.37 |
> | **Ant-Rand-Dir** | 579.03 | -63.79 | 92.05  |
>
> The result shows that putting noises to the first few query feedbacks would largely degrade the performance of baselines. In comparison, the performance drop of ANOLE is much more modest, which demonstrates its outstanding robustness.
>
> [2] Lee, K., Smith, L., Dragan, A., & Abbeel, P. (2021, June). B-Pref: Benchmarking Preference-Based Reinforcement Learning. In *Thirty-fifth Conference on Neural Information Processing Systems Datasets and Benchmarks Track (Round 1)*.

---

> > ### Comment · Reviewer_L5bT · 2022-08-07
> > **Response**
> >
> > I really appreciate the detailed responses from the authors.
> >
> > I change the score as "weak accept" since the authors handled my concerns on (1) baseline, (2) new tasks and (3) difference noise model. If the paper is accepted, it would be nice if the authors could also include real human experiments.

---

> > > ### Author Response · Authors · 2022-08-09
> > > **Thanks for the inspiring review and constructive suggestions.**
> > >
> > > Thanks for the inspiring comments and for increasing the score. We are happy that most of the concerns are now addressed.
> > >
> > > In addition to the previous response, we would like to present some preliminary experiments that use real human feedback to perform meta-test adaptation. The results show that the learned meta-adaptation policy is compatible with human preference (see the following table).
> > >
> > > | Human Mode       | ANOLE  | Greedy | Random |
> > > | ---------------- | ------ | ------ | ------ |
> > > | **Ant-Fwd-Back** | 935.33 | 922.30 | 924.72 |
> > > | **Ant-Rand-Dir** | 500.89 | 495.57 | 447.56 |
> > >
> > > The observation is similar to that of the Boltzman preference model. The performance gap between ANOLE and baselines is slighter than the gap observed in the uniform-noise setting. That is because, for these testing environments, real human feedbacks are more accurate than the uniform-noise oracle in the average statistics. We will continually work on it and seek for an appropriate experiment setting to capture real human errors beyond just evaluating with human interaction. In addition, we will pay special attention to checking the conference guideline for human-subject experiments to make sure the experiment setting is well controlled.

---

### Official Review · Reviewer_rdwj · 2022-07-11

**Rating:** 7
**Confidence:** 4
**Soundness:** 3 good
**Presentation:** 3 good
**Contribution:** 2 fair

**Summary:**

This work considers a new few-shot meta-RL setting, where the agent does not observe rewards during its few shots on a new task, but instead submits pairs of trajectories to a preference oracle, which returns which of the two trajectories is preferred (i.e., would achieve higher returns). The idea is that querying a preference oracle may require less supervision than obtaining reward labels on these trajectories. Then, to solve this setting, this work proposes a new method for task inference, where a preference predictor is learned to imitate the preference oracle, and then the task is inferred by performing an error-tolerant form of binary search with the preference predictor: the idea is to find a task under which the preference predictor most agrees with the preference oracle. This work evaluates the proposed method on standard MuJoCo meta-RL tasks, where only the reward function changes between tasks, and finds that the proposed method outperforms vanilla PEARL (which is not designed to handle this setting). Furthermore, the experiments show that adding error tolerance to the binary search procedure is important when the oracle includes noise.

**Questions:**

I've condensed the main questions in order of most important to least important I have from my review above here:
- When is preference-based meta-RL practically better than the standard meta-RL setting (and vice versa)? How does the amount of supervision compare between preference-based meta-RL and standard meta-RL?
- How do the various methods compare in terms of feedback efficiency? How does performance change as a factor of $K$?
- Are there reasonable ways to compare with other existing meta-RL algorithms?
- What happens when $K_E$ is mis-specified?

**Limitations:**

This work could benefit from more discussion about the limitations of the proposed new setting. Additionally, this work restricts itself to the setting where only the rewards change between tasks, while existing meta-RL works often also consider changing dynamics. It would be useful to see some discussion about whether this restriction is fundamental to the proposed method or setting, or if it can be relaxed in some form.

**Strengths And Weaknesses:**

# Originality
This work considers a novel setting, where rewards are not observed in the few shot episodes of meta-RL, and instead, only preferences between pairs of trajectories are given. In addition, and as a result of this novel setting, this work offers a new connection with Renyi-Ulam's games and the notion of Berlekamp's volume, and is overall quite original. However, it is worth noting that the introduction slightly overclaims: Line 48 states that "reward-free few-shot policy adaptation" has not been explored in prior work, while [1] does reward-free few-shot policy adaptation, though in quite a different way.


# Quality
- My main concern about the quality is that this work does not obviously validate its claims about feedback efficiency. This work states in several places that it aims to design an algorithm that meets two goals (1) feedback efficiency, and (2) error tolerance. The first goal seems particularly important, since that is a key motivation behind the setting: if it actually requires more supervision than reward labels to provide, then it's unclear why we should consider a preference-based setting. However, the experiments neither test whether the proposed method provides feedback efficiency compared to other methods in this setting, nor whether the preference-based setting requires less supervision than a reward-based setting. I believe that both are critical.
- Another concern that I have with the experiments is that the experiments consider the case where $K_E$ is correctly specified. However, in practice, it seems unlikely that the noise level of the black-box preference oracle is known a priori. It therefore also seems important to investigate how performance changes if $K_E$ does not match the true noise level.
- A final concern with the experiments is the set of baselines. Since this work considers a new setting, it is natural that many prior works do not directly apply to the setting, and therefore finding appropriate baselines is tricky. However, while PEARL is an important meta-RL approach, there is a rich set of other meta-RL methods (e.g., RL^2 [2], VariBAD [3]), and it would be helpful to understand whether other existing meta-RL methods also work well in this setting. I note that it's unclear to me how to directly adapt these methods to the new setting in this work, so I do not hold this against this work, but it would also be nice to compare with other meta-RL algorithms beyond PEARL.

# Clarity
This work is generally understandable, although the presentation is confusing in some places and requires reading ahead and re-reading. I've noted a few areas below:
- Lines 163-177 discuss how the preference predictor helps transform the setting to Renyi-Ulam's Game by transforming binary preferences into subset inclusion. However, how the preference predictor achieves this is not clear until later in the binary search discussion. In the introduction to the preference predictor, it's only clear that the preference predictor is learned to match the black-box preference oracle.
- In addition, while the connection to Renyi-Ulam's Game is interesting, introducing it while introducing the main ideas of the algorithm makes it more challenging to understand what's going on. Perhaps first introducing the main ideas of the algorithm, and later showing how they connect to Renyi-Ulam's Game would be easier to understand.
- The notation can be quite difficult to parse in some places, with many superscripts. It would be nice if this could be simplified in some areas, if possible.

# Significance
- My main concern with this work's significance is how practically relevant the setting is. As I alluded to in my section on quality, it's unclear to me that the preference-based setting indeed requires less supervision than providing reward labels. In fact, in the domains included in the experiments, it seems like providing reward labels requires the same amount of supervision. Computing the binary preference seems to require the same computations as the reward labels, e.g., computing the distance from the goal, direction, or desired velocity. I would appreciate if the authors could comment on settings where it is clear that preference-based meta-RL is preferably to the standard setting.

[1] Explore then Execute: Adapting without Rewards via Factorized Meta-Reinforcement Learning. Evan Zheran Liu, Aditi Raghunathan, Percy Liang, Chelsea Finn. https://openreview.net/pdf?id=La1QuucFt8-

[2] RL2: Fast Reinforcement Learning via Slow Reinforcement Learning. Yan Duan, John Schulman, Xi Chen, Peter L. Bartlett, Ilya Sutskever, Pieter Abbeel. https://arxiv.org/abs/1611.02779

[3] VariBAD: A Very Good Method for Bayes-Adaptive Deep RL via Meta-Learning. Luisa Zintgraf, Kyriacos Shiarlis, Maximilian Igl, Sebastian Schulze, Yarin Gal, Katja Hofmann, Shimon Whiteson. https://openreview.net/forum?id=Hkl9JlBYvr

---

> ### Author Response · Authors · 2022-08-02
> **Response to Reviewer rdwj**
>
> Thanks for the comments. We provide clarification to your questions and concerns as below. If our response does not fully address your concerns, please post additional questions and we will be happy to have further discussions.
>
> We note that the reviewer's major concern is about the position of preference-based meta-RL in the general RL community. To address this question, we first discuss this issue in the general literature background (see Q1.1) and then move on to detailed clarifications specific to this paper (see Q1.2 and Q1.3).
>
> **Q1.1: How relevant the setting is?**
>
> The relation between preference-based meta-RL and standard meta-RL setting is similar to their single-task counterparts, *i.e.*, preference-based RL *vs.* standard RL. Preference-based RL is a sub-area of human-in-the-loop RL. The purpose of utilizing human supervision is not only to exploit the reward engineering problem with human experts [1] but also to promote human-robot interaction when serving non-expert users [2,3]. We explain these two aspects as follows:
>
> 1. As an RL problem, the major characteristic of preference-based RL is the usage of trajectory-wise binary preference labels instead of step-wise numerical rewards. The information carried by trajectory-wise binary preference labels is strictly less than step-wise numerical rewards (see Q1.2 for details). Designing specific RL algorithms for preference-based supervision is a long-lasting classical problem for more than ten years [4].
>
> 2. Another goal of preference-based methods is to develop user-friendly policies for non-expert human users. Using a preference-based policy, the user is able to customize the policy behavior by answering binary preference queries, which is an easy way to configure the user profile. In this paper, we develop the setting of preference-based meta-RL to enhance the impact of preference-based RL on this aspect. As mentioned by reviewer kW96, developing effective preference-based meta-RL algorithms may help human users to better communicate their preferences with the deployed AI agents.
>
> [1] Wirth, C., Akrour, R., Neumann, G., & Fürnkranz, J. (2017). A survey of preference-based reinforcement learning methods. *Journal of Machine Learning Research, 18*(136), 1-46.
>
> [2] Allen, T. E., Chen, M., Goldsmith, J., Mattei, N., Popova, A., Regenwetter, M., ... & Zwilling, C. (2015, September). Beyond theory and data in preference modeling: Bringing humans into the loop. In *International Conference on Algorithmic Decision Theory* (pp. 3-18). Springer, Cham.
>
> [3] Prewett, M. S., Johnson, R. C., Saboe, K. N., Elliott, L. R., & Coovert, M. D. (2010). Managing workload in human-robot interaction: A review of empirical studies. *Computers in Human Behavior, 26*(5), 840-856.
>
> [4] Akrour, R., Schoenauer, M., & Sebag, M. (2011, September). Preference-based policy learning. In *Joint European Conference on Machine Learning and Knowledge Discovery in Databases* (pp. 12-27). Springer, Berlin, Heidelberg.
>
> **Q1.2: How does the amount of supervision compare between preference-based meta-RL and standard meta-RL?**
>
> From the viewpoint of developing RL algorithms, the information carried by trajectory-wise binary preference labels is strictly less than step-wise numerical rewards. It is straightforward that step-wise numerical rewards can simulate a trajectory-wise preference order. However, converting trajectory-wise binary preference to step-wise numerical rewards is a centric challenge of preference-based RL, especially when the preference feedbacks carry some extent of noise [5].
>
> In the context of standard meta-RL, the step-wise reward function plays an important role for the meta-RL agent to infer the task goal. *e.g.*, PEARL uses a context encoder to infer the task variable from reward signals. During meta-testing, a PEARL agent requires the access to step-wise rewards as what is used in meta-training. More specially, for PEARL-like adaptation protocol, the reward function used in meta-testing should follow the same distribution as what is trained in meta-training (*i.e.*, the scale of reward signals is also important), which is a constraint for practical use. In comparison, preference-based meta-RL only assumes a partial order between trajectories to perform meta-test adaptation, which requires less supervision during meta-testing.
>
> [5] Lee, K., Smith, L., Dragan, A., & Abbeel, P. (2021, June). B-Pref: Benchmarking Preference-Based Reinforcement Learning. In *Thirty-fifth Conference on Neural Information Processing Systems Datasets and Benchmarks Track (Round 1)*.

---

> > ### Author Response · Authors · 2022-08-02
> > **Response to Reviewer rdwj (cont'd)**
> >
> > **Q1.3: When is preference-based meta-RL practically better than the standard meta-RL setting (and vice versa)? Are there reasonable ways to compare with other existing meta-RL algorithms?**
> >
> > When (1) a step-wise dense reward function is accessible during meta-testing, and (2) the policy quality is the only evaluation metric, the standard meta-RL algorithms are preferable to preference-based meta-RL. The main purpose of preference-based meta-RL is not to outperform standard meta-RL in the standard setting.
> >
> > As we discussed in Q1.1, the main purpose of preference-based meta-RL is to serve non-expert human users. In the case when the user cannot specify a dense reward function, a preference-based meta-RL agent would be preferable to a standard one.
> >
> > **Q2.1: How do the various methods compare in terms of feedback efficiency?**
> >
> > The comparison of feedback efficiency contains two aspects:
> >
> > 1. Comparison to standard preference-based RL. In our experiment setting, the meta-policy of ANOLE is able to perform meta-test adaptation with no more than 10 preference feedbacks. We would like to remark on its high feedback efficiency compared to training policies from scratch, which usually costs hundreds or thousands of preference feedbacks [4]. We emphasize the feedback-efficiency gap between the meta and the non-meta counterparts of preference-based RL, *i.e.*, using less amount of feedbacks to produce meaningful policies.
> >
> > 2. Comparison to baselines of preference-based meta-RL. We limit the interaction budget to 10 preference feedbacks, since it is a sufficiently small amount. All algorithms are competing using the same amount of feedbacks. More specifically, we compare the effectiveness to utilize the same feedback budget, which is another quantification of feedback efficiency, *i.e.*, using the same feedback budget to produce better policies.
> >
> > [4] Christiano, P. F., Leike, J., Brown, T., Martic, M., Legg, S., & Amodei, D. (2017). Deep reinforcement learning from human preferences. *Advances in Neural Information Processing Systems, 30*.
> >
> > **Q2.2: How does performance change as a factor of $K$?**
> >
> > In Appendix D, we present the performance of meta-policy at each adaptation step. The experiment can be concluded by three observations:
> >
> > 1. In general, the performance improvement gained by each new feedback decays along with the interactive query procedure.
> >
> > 2. For baselines that are not designed for error tolerance, only the first $k\leq 4$ feedbacks can improve the average return. That is because these baselines do not have error-tolerating modules, and thus their belief on the task variable will rapidly converge (maybe reach a wrong decision).
> >
> > 3. In comparison to baselines, the policy quality of ANOLE can continually improve with the increase of the number of feedbacks. That is because ANOLE can use new feedbacks to denoise previous feedbacks.
> >
> > **Q3: What happens when $K_E$ is mis-specified?**
> >
> > In Appendix C, we evaluate the performance of ANOLE and baselines under different magnitudes of oracle noises, *i.e.*, the error probability $\epsilon\in\{0.0, 0.1, 0.2, 0.3\}$ where $\epsilon=0.2$ corresponds to the default setting. The result shows that, with the noise magnitude increasing, the gap between ANOLE and baselines becomes larger. The performance of ANOLE changes smoothly with the increase of noise magnitude, which indicates $K_E$ is not a hard threshold.
> >
> > **Q4: It is worth noting that the introduction (Line 48) slightly overclaims.**
> >
> > Thanks for introducing the wonderful related work. We refined the discussion around line 48 and cited the referred paper. In the next revision, we will pay special attention to refining our paper presentation.

---

> > > ### Comment · Reviewer_rdwj · 2022-08-08
> > > **Thank you for the response!**
> > >
> > > Thanks for the detailed response, which have helped alleviate my concerns, particularly on the experimental side. I support accepting this paper and will increase my score.
> > >
> > > There are still a few aspects that I think could be strengthened, if addressed:
> > > - Part of my concern that remains unaddressed is that I understand that in general, receiving binary preferences between two trajectories represents less supervision than receiving reward labels. However, my concern is that in the provided experiments, it's not clear to me that it really does require less supervision. For example, in Ant-Goal, computing the binary feedback between two trajectories involves computing the distance between the end states of the two trajectories and the goal. However, if it is possible to do this already, it seems like computing the distance between the current state and the goal at every timestep is not that challenging. I understand that these MuJoCo benchmarks are fairly common in the meta-RL literature, and so I emphasize that I still support acceptance. But I think the experimental domains could be more compelling on domains where gathering preferences is more realistic.
> > > - Along these lines, the supervision tradeoff between binary preferences and reward labels is not so clear cut. For example, some meta-RL algorithms are capable of solving new tasks after only 1 episode on the task. It is not immediately obvious whether 1 episode with reward labels constitutes more supervision than 10 binary preference labels, particularly if the episodic rewards are sparse. I am definitely aware of situations where the 10 binary preference labels are preferable, but I think this warrants further discussion as well.
> > > - The author response primarily discusses feedback efficiency during meta-testing, which is notably fairly efficient. However, if I am not misunderstanding, the algorithm still requires an enormous amount of feedback during meta-training. I **do not** hold this against this work, as it is typical that meta-RL algorithms require significant samples during meta-training. But I think this work could add discussion and transparency about this aspect.
> > >
> > > To be clear, I think this work already is above the bar, but I think the inclusion of these aspects could strengthen the paper.

---

> > > > ### Author Response · Authors · 2022-08-09
> > > > **Thanks for the inspiring review and constructive suggestions.**
> > > >
> > > > Thanks for the inspiring comments and for increasing the score. The review gives many constructive suggestions and indicates many valuable problems.
> > > >
> > > > - **The correspondence between the preference order and the dense reward function.** The reviewer's understanding is correct. In the current experiment benchmark, the preference order over trajectories can be well represented by a trivial dense reward function, which cannot fully reveal the specific characteristics of preference-based learning. This is also a common limtation of related works and is worth further studies. In general, characterizing the correspondence between the preference order and the dense reward function is a long-lasting problem of preference-based RL. A recent theoretical work shows that there exists some preference partial orders cannot be represented by step-wise rewards [1]. The current work (including related works) have not built the connection between these theoretically hard cases and practical scenarios, and thus it deserves special attention for future works.
> > > >
> > > > - **The sample cost of meta-RL.** We agree with the reviewer that the large sample cost is a critical bottleneck of meta-RL algorithms. Regarding this issue, we would like to discuss a valuable property of ANOLE. As mentioned in section 4.1, ANOLE is an adaptation module and can be built upon any meta-RL or multi-task RL algorithms with latent policy encoding, *i.e.*, the meta-training and meta-testing procedures can be well decoupled. A promising future direction is developing preference-based meta-adaptation upon other more sample-efficient meta-training modules, such as meta-training from demonstrations [2] and unsupervised skill discovery [3], which may serve an effective way to improve the efficiency of meta-training.
> > > >
> > > > We note that the conference allows an additional content page for the camera-ready version of papers. We may have a great chance to enrich the discussions of the literature background, the limitations of the current work, and some promising future works in the revision upon acceptance.
> > > >
> > > > [1] Abel, D., Dabney, W., Harutyunyan, A., Ho, M. K., Littman, M., Precup, D., & Singh, S. (2021). On the expressivity of markov reward. *Advances in Neural Information Processing Systems, 34*, 7799-7812.
> > > >
> > > > [2] Lynch, C., Khansari, M., Xiao, T., Kumar, V., Tompson, J., Levine, S., & Sermanet, P. (2020, May). Learning latent plans from play. In *Conference on robot learning* (pp. 1113-1132). PMLR.
> > > >
> > > > [3] Gupta, A., Eysenbach, B., Finn, C., & Levine, S. (2018). Unsupervised meta-learning for reinforcement learning. In *International Conference on Learning Representations*, 2020.

---

### Official Review · Reviewer_kW96 · 2022-07-11

**Rating:** 7
**Confidence:** 4
**Soundness:** 3 good
**Presentation:** 3 good
**Contribution:** 3 good

**Summary:**

The paper studies the setting of meta-reinforcement learning, but with the constraint that rewards are not observable at adaptation time. Rather, the algorithm must infer the task by querying a noisy oracle (e.g. a human) which indicates its preferences by ranking given pairs of trajectories. The goal is to adapt to the test task with few queries, to minimize burden on the human.

The authors draw a connection between preference-based adaptation and Rényi-Ulam’s game, an information-theoretic game in which one player tries to guess what the other player is thinking, but they can only communicate via yes/no questions over a noisy channel. Based on this, they propose to use a classical uncertainty quantification tool known as Berlekamp’s volume in order to choose which queries to propose to the oracle. Specifically, they choose the query that maximizes the worst-case uncertainty reduction.

Experimental results indicate that the use of Berlekamp’s volume provides improvement over baselines such as using greedy queries which do not consider the possibility of noise in the oracle’s responses, as well as modifications of PEARL to the preference-based adaptation setting.

**Questions:**

The query trajectories in Eq. (6) are selected from “the experience buffer”. Based on the fact that you have to subsample this buffer (presumably for computational purposes), I assume this buffer includes trajectories from meta-training time. Is it reasonable to assume that those trajectories will be available at adaptation time?

**Limitations:**

The authors have addressed limitations, namely that their method cannot completely eliminate the effect of a noisy oracle (which is to be expected).
I agree with the authors that the risk of negative societal impacts is negligible. If anything, the impact will be positive because it improves the ability of humans to communicate their preferences to deployed AI agents.

**Strengths And Weaknesses:**

Strengths:
* The setting of adaptation without dense rewards is useful because it meta-RL more practically applicable.
* The connection between preference-based adaptation in meta-RL and Rényu-Ulam’s game is novel to my knowledge, and brings information-theoretic tools (specifically, the Berlekamp volume) to bear on the design of the algorithm. This is a useful contribution.
* The experiments demonstrate improvement due to use of the BVol-based query proposal.
* The paper is well-written, being organized, clear, and generally well-motivated.

Weaknesses:
* The paper would benefit from a bit more explanation of the formula for Berlekamp’s volume, since most readers will not be familiar with it.
* Another useful comparison would be ordinary PEARL using the dense rewards. (I guess you do not even have to re-run it; just use their published results.) This essentially serves as an (unattainable) upper bound on the performance we could expect with a PEARL-like adaptation algorithm, and the gap would show how much is lost by using the preference-based feedback.

---

> ### Author Response · Authors · 2022-08-02
> **Response to Reviewer kW96**
>
> Thanks for the comments. We provide clarification to your questions and concerns as below. If our response does not fully address your concerns, please post additional questions and we will be happy to have further discussions.
>
> **Q1: Regarding the experience buffer.**
>
> In our implementation, we sample trajectory pairs from the experience buffer and compute Berlekamp's volume to determine the query pair. This setting mimics a practical scenario where the user fills in an online form [1] with binary questions to configure his/her preference. *e.g.*, the user watches some trajectory videos and labels which one he/she prefers. We consider this setting since it leads to simple implementation and makes the evaluation procedure more computationally efficient.
>
> In addition, it may be a promising future direction to design a more meticulous query generation algorithm that generates informative query trajectories instead of simply sampling candidates from the experience buffer. In the literature of meta-RL, some previous works [2-3] use a separate exploration policy to collect the context information for meta-test adaptation. These ideas can be further combined with the framework of ANOLE to benefit preference-based meta-RL.
>
> [1] Sumitkumar, K., Sheetal, G., & Debajyoti, M. (2014). User Profiling Trends, Techniques and Applications. *International Journal of Advance Foundation and Research in Computer (IJAFRC) Volume, 1*.
>
> [2] Liu, E. Z., Raghunathan, A., Liang, P., & Finn, C. (2021, July). Decoupling exploration and exploitation for meta-reinforcement learning without sacrifices. In *International Conference on Machine Learning* (pp. 6925-6935). PMLR.
>
> [3] Zhang, J., Wang, J., Hu, H., Chen, T., Chen, Y., Fan, C., & Zhang, C. (2021, July). Metacure: Meta reinforcement learning with empowerment-driven exploration. In *International Conference on Machine Learning* (pp. 12600-12610). PMLR.
>
> **Q2: More explanation for Berlekamp's volume.**
>
> Berlekamp's volume is a special tool in the literature of information theory, which characterizes the inherent uncertainty of the communication channel. The values of Berlekamp's volume can be roughly interpreted as the number of valid future outcomes given the current state of information. For more technical details, We would like to guide the readers to a classical paper [4], which uses a stroy-talking tone to illustrate the construction of Berlekamp's volume.
>
> We note that the conference allows an additional content page for the camera-ready version of papers. We may have a great chance to enrich the discussions of the literature background of Berlekamp's volume and Rényi-Ulam's game in the revision upon acceptance.
>
> [4] Lawler, E. L., & Sarkissian, S. (1995). An algorithm for “Ulam's Game” and its application to error correcting codes. *Information processing letters, 56*(2), 89-93.
>
> **Q3: Comparison to ordinary PEARL using the dense rewards.**
>
> Thanks for the advice on improving our paper presentation. We updated Figure 1 in the paper content to include a bar that indicates the performance of dense-reward PEARL. In the next revision, we will add more descriptions and discussions regarding the comparison to dense-reward PEARL.

---

### Official Review · Reviewer_ZtTs · 2022-07-13

**Rating:** 6
**Confidence:** 4
**Soundness:** 3 good
**Presentation:** 3 good
**Contribution:** 3 good

**Summary:**

This paper proposes an approach to incorporate preference feedback for meta-RL, as opposed to assuming access to dense environment rewards. This takes the form of querying the oracle for ranking two trajectories, and the method leverages ideas from information theory (berlekamp volume) to deal with noise in oracle feedback. The method is evaluated on 6 gym meta-RL benchmark tasks.


**Questions:**

-> Can this approach actually be used to train agents with a human oracle (without using the mujoco simulator to rank trajectories?). Experiments using human feedback like in section 5.4 of [2] will help demonstrate the efficacy and practicality of the proposed approach.

-> Does the approach scale to more complex meta-learning benchmarks, such as meta-world?


[2] : Feedback-Efficient Interactive Reinforcement Learning via Relabeling Experience and Unsupervised Pre-training (Lee et al.)



**Limitations:**

Limitations are adequately discussed

**Strengths And Weaknesses:**

Significance -

-> The paper addresses an important limitation for practical application of meta-RL, i.e the assumption of dense environment rewards. Querying an oracle (eg: human user) is much more feasible. However, in the included experiments, the number of samples required for meta-training is on the order of a million, and the number of preference queries per step is around 10, putting the total number of queries required on the order of 10 million, so training with this scheme would still be impractical for deployment. It is nonetheless a step in the right direction, since dense rewards are very difficult to specify in the real world for arbitrary tasks, while ranking trajectories can still be done. Further since current approaches to meta-RL don’t commonly use preference feedback, this paper will be valuable to the community in helping to start work in this direction.

Originality -

-> The method proposed involves bringing preference based learning to meta-RL. The authors leverage ideas from information theory (Berlekamp volume) to account for a noisy oracle, and add this to standard context based meta-RL. This formulation seems novel and while the performance benefit over a simpler greedy binary search baseline is marginal, this paper might inspire subsequent work that borrows from similar ideas.

-> The paper shows advantages of using non-parametric latents (instead of using an inference network like [1]) for the binary feedback problem, as even a random-query strategy can outperform a baseline based on [1].

-> The related work adequately discusses context based meta-RL and preference based learning.

[1] : Efficient Off-Policy Meta-Reinforcement Learning via Probabilistic Context Variables (Rakelly et al.)


Clarity -

-> The paper is clearly written, well motivated and easy to follow, and contains ample details (algorithm boxes, hyperparameter details) to aid in reproducibility.

Quality -

-> The preference based meta-RL formulation seems sound. While the experiments only include gym domains these are standard meta-Rl benchmarks. The authors also include ablations where increase in oracle noise leads to worse performance, as expected.

---

> ### Author Response · Authors · 2022-08-02
> **Repsonse to Reviewer ZtTs**
>
> Thanks for the comments. We provide clarification to your questions and concerns as below. If our response does not fully address your concerns, please post additional questions and we will be happy to have further discussions.
>
> **Q1: Experiments with a human oracle.**
>
> We conduct experiments that use real human feedback to perform meta-test adaptation. The results in terms of average return values are presented in the following table.
>
> | Human Mode       | ANOLE  | Greedy | Random |
> | ---------------- | ------ | ------ | ------ |
> | **Ant-Fwd-Back** | 935.33 | 922.30 | 924.72 |
> | **Ant-Rand-Dir** | 500.89 | 495.57 | 447.56 |
>
> We note that, for these testing environments, human feedbacks are more accurate than the uniform-noise oracle considered in the paper, and thus the gap between ANOLE and baselines becomes slighter. That is because the generated query trajectory pairs can usually be clearly compared. The performance of ANOLE and greedy/random query strategies are comparable since errors hardly occur.
>
> Regarding this result, we would like to emphasize that the main purpose of ANOLE is to tolerate unintended errors of human users. More specifically, a robust algorithm is expected to tolerate a few amounts of irrational errors. To support this point, we design another noisy oracle, called "Hack Mode", which always gives wrong feedbacks for the first 20\% queries and keeps correct for the remaining 80\%. This noise mode is designed to hack search-based query strategies, since the first few queries are usually the most informative.
>
> | Hack Mode        | ANOLE  | Greedy | Random |
> | ---------------- | ------ | ------ | ------ |
> | **Ant-Fwd-Back** | 945.41 | 433.52 | 652.37 |
> | **Ant-Rand-Dir** | 579.03 | -63.79 | 92.05  |
>
> The result shows that putting noises to the first few query feedbacks would largely degrade the performance of baselines. In comparison, the performance drop of ANOLE is much more modest, which demonstrates its outstanding robustness.
>
> **Q2: Experiments on Meta-World manipulation tasks.**
>
> We conduct experiments on two Meta-World manipulation tasks. We present the average return values of ANOLE and baselines in the following table.
>
> | Meta-World Benchmark  | ANOLE | Greedy | Random |
> | --------------------- | ----- | ------ | ------ |
> | **ML1-Push-v2**       | 89.19 | 53.75  | 48.35  |
> | **ML1-Pick-Place-v2** | 8.96  | 8.97   | 8.93   |
>
> These results are recorded when training $10^7$ environment steps. We note that, in the Meta-World benchmark paper [1], PEARL requires $3\times 10^8$ environment steps to achieve a remarkable manipulation success rate in these two tasks. Our computation resource cannot support us to complete the whole training procedure during the short rebuttal period. We will continue to work on it.
>
> [1] Yu, T., Quillen, D., He, Z., Julian, R., Hausman, K., Finn, C., & Levine, S. (2020, May). Meta-world: A benchmark and evaluation for multi-task and meta reinforcement learning. In *Conference on Robot Learning* (pp. 1094-1100). PMLR.
>
> **Q3: Regarding the training cost.**
>
> The review states "training with this scheme would still be impractical for deployment". The training cost is a fundamental challenge for meta-RL. To achieve fast adaptation, it is reasonable that the meta-training procedure requires more samples than single-task training.
>
> Regarding this issue, we would like to discuss a valuable property of ANOLE. As mentioned in section 4.1, ANOLE is an adaptation module and can be built upon any meta-RL or multi-task RL algorithms with latent policy encoding, *i.e.*, the meta-training and meta-testing procedures can be well decoupled. A promising future direction is developing preference-based meta-adaptation upon other more sample-efficient meta-training modules, such as meta-training from demonstrations [2] and unsupervised skill discovery [3], which may serve as an effective way to improve the efficiency of meta-training.
>
> [2] Lynch, C., Khansari, M., Xiao, T., Kumar, V., Tompson, J., Levine, S., & Sermanet, P. (2020, May). Learning latent plans from play. In *Conference on robot learning* (pp. 1113-1132). PMLR.
>
> [3] Gupta, A., Eysenbach, B., Finn, C., & Levine, S. (2018). Unsupervised meta-learning for reinforcement learning. In *International Conference on Learning Representations*, 2020.

---

> > ### Comment · Reviewer_ZtTs · 2022-08-10
> > **Rebuttal Response**
> >
> > I thank the authors for their response. The new experiments with human feedback and the metaworld domains definitely strengthen the claims in the papers, and further show where the method is most effective - in dealing with noisy feedback (since with noiseless feedback the gap between anole and a random querying strategy is very small). Nonetheless the extent of new experiments with human oracle is not large enough for me to increase score given that I'm already in favor of accepting this paper, and so I will not be changing my score.

---

> > > ### Author Response · Authors · 2022-08-10
> > > **Thanks for the inspiring review and constructive suggestions.**
> > >
> > > Thanks for your responsive reply and for the inspiring review that helps us to improve our work. We will continually work on human-based experiments and incorporate rebuttal discussions into the next revision.

---

### Meta-Review · Area_Chair_M5LA · 2022-08-20

**Recommendation:** Accept
**Confidence:** Certain

**Metareview:**

**Strengths**: This paper introduces an interesting new problem setting (meta-RL for preference-based adaptation) that is of practical relevance, along with a sensible new approach that makes progress on addressing the problem.

**Weaknesses**: After the author discussion period, there are two remaining concerns:
* Additional experiments, particularly human experiments and more complex tasks.
* Further discussion / clarification / support for when preference-based feedback is preferable to other forms of supervision, e.g., sparse rewards.

Overall, the reviewers and AC agree that this paper makes a worthy contribution to NeurIPS, despite the weaknesses. Nonetheless, we expect that human experiments in particular would help with both of the weaknesses and increase the impact of the paper, so we especially encourage the authors to work on such experiments before the camera ready version.

**Award:**

No

---

### Decision · Program_Chairs · 2022-09-14

Accept